# Warm rings in mesoscale eddies in a cold straining ocean

Huizi Dong [1,2,3] ✉, Meng Zhou [1,2,3] ✉, James C. McWilliams[4], Roshin P. Raj [5], Francesco d'Ovidio [6], Ilker Fer [7], Lixin Qu[1,2,3], Bo Qiu[8], Lia Siegelman [9], Zhengguang Zhang [10,11], Walker O. Smith Jr [1,2,3] & Ann Kristin Sperrevik [12]

The warm and saline Atlantic Water has long been recognized as being subjected to substantial heat loss during its transit towards the polar regions. In particular, the Lofoten Basin, a subpolar sea with energetic eddy activity and strong air-sea interactions, plays a crucial role in the transformation of Atlantic Water. Vertical heat transport at submesoscales (0.1-10 km) in the Lofoten Basin is potentially a key link in the heat transfer to the atmosphere. Here, based on multi-year Seaglider observations augmented by satellite altimeters, radiometers, and high-resolution numerical model results, we evaluate the oceanic vertical heat transport in the Lofoten Basin and demonstrate how geostrophic strain enhances heat transport. The enhancement is found to be associated with submesoscale ageostrophic motions along the mesoscale eddy edges, occurring on spatial scales smaller than 10 km and below the mixed layer depth. These strain-induced submesoscale vertical motions transport heat from the ocean interior to the surface, leading to a 0.4 °C increase in sea surface temperature and the formation of "warm ring" structures in both cyclones and anticyclones. The dominant role of submesoscale heat transport likely represents the primary mechanism for substantial heat loss from Atlantic Water in the Lofoten Basin.

The Lofoten Basin (LB) is the largest oceanic heat reservoir in the Nordic Seas and a region with strong air-sea interactions[1,2]. The North Atlantic Current is an important part of the oceanic conveyor belt that transports heat from the equator to the high latitudes[3]. As it flows through the Nordic Seas, especially the Lofoten Basin, the Atlantic Water undergoes transformation driven by both substantial heat loss to the atmosphere[4,5] and lateral heat transport through mesoscale eddies[6,7]. This transformation significantly impacts the hydrographic environment, sea ice cover, and ecosystems in the region and ultimately the Arctic Ocean[8]. The LB Fig. 1a) is characterized by a distinct shelf-slope-basin topography, which largely determines the general circulation pattern, water mass exchange, formation of eddies and transport of heat and nutrients in the region[9,10]. The LB hosts energetic mesoscale eddy fields, with a large number of eddies shedding from

[1]Key Laboratory of Polar Ecosystem and Climate Change, Ministry of Education and School of Oceanography, Shanghai Jiao Tong University, Shanghai, China. [2]Shanghai Key Laboratory of Polar Life and Environment Sciences, Shanghai Jiao Tong University, Shanghai, China. [3]Shanghai Frontiers Science Center of Polar Science, Shanghai Jiao Tong University, Shanghai, China. [4]Department of Atmospheric and Oceanic Sciences, University of California, Los Angeles, CA, USA. [5]Nansen Environmental and Remote Sensing Center, Norway and Bjerknes Center for Climate Research, Bergen, Norway. [6]Sorbonne Université, CNRS, IRD, MNHN, Oceanography and Climate Laboratory: Experiments and Numerical Approaches (LOCEAN-IPSL), Paris, France. [7]Geophysical Institute, University of Bergen and Bjerknes Center for Climate Research, Bergen, Norway. [8]Department of Oceanography, University of Hawaii at Manoa, Honolulu, HI, USA. [9]Scripps Institution of Oceanography, University of California, San Diego, La Jolla, CA, USA. [10]Key Laboratory of Physical Oceanography, Frontier Science Center for Deep Ocean Multispheres and Earth System (FDOMES), Ocean University of China, Qingdao, China. [11]Laoshan Laboratory, Qingdao, China. [12]Division for Ocean and Ice, Norwegian Meteorological Institute, Oslo, Norway. ✉e-mail: huizidong@sjtu.edu.cn; meng.zhou@sjtu.edu.cn

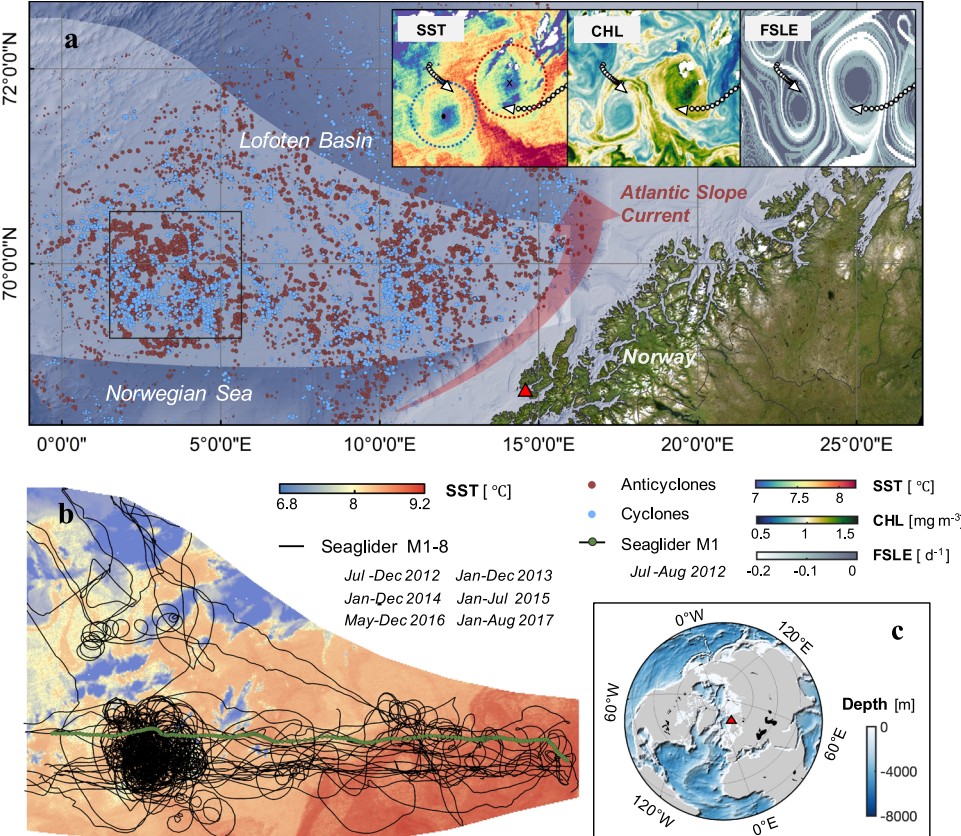

**Fig. 1 | Distribution of cyclonic and anticyclonic eddies in the Seaglider sampling region. a** Map of Lofoten Basin region with the locations of all identified cyclonic (blue dots) and anticyclonic (red dots) eddies observed during eight Seaglider missions between July 2012 and August 2017. The light gray shaded area corresponds to the region covered by Seaglider observations. The three insets in (**a**) show the instantaneous sea surface temperature (SST), chlorophyll (CHL) and finite-size Lyapunov exponent (FSLE) fields as the glider sg559 M1 traversed a pair of eddies within the black boxed regions on 8 June 2017. The white dots and white arrows represent the trajectory and direction of the glider, respectively. **b** Trajectories of eight Seaglider missions between 4 July 2012 and 9 August 2017 (black lines) superimposed on a snapshot of satellite-based SST. The trajectory with green dots represents the first section of the M1 mission between 6 July 2012 and 16 August 2012. The location of the Lofoten Islands in the study area is marked by red triangles in panels (**a**) and (**c**).

the most unstable area (near the Lofoten Islands)[11,12] of the Norwegian Atlantic Slope Current and propagating westward, feeding the quasi-permanent Lofoten Basin Eddy (LBE), which subsequently is maintained by bottom topography[1,13,14]. Regional studies have demonstrated that mesoscale eddies and other lateral processes (e.g., advection and mixing) transport heat towards the LB, playing an important role in the transformation of Atlantic Water[5,6,11]. During winter, there is an increase in lateral exchanges and heat flux between the permanent anticyclonic LBE and its surroundings, primarily driven by instabilities and frequent eddy merging events[4,6,15]. Vertical heat exchange has also been reported to be significantly influenced by processes such as double-diffusive convection and internal-wave-driven mixing in polar regions[16,17]. Over the past decade, dynamical studies have increasingly highlighted the crucial role of submesoscale motions in facilitating vertical exchanges of heat and nutrients between the surface and the deep ocean[18–20]. However, their influence on the vertical heat transport (VHT) in the eddy-rich Atlantic Water transformation zone has received limited attention due to a lack of high-resolution observations.

To address this knowledge gap in the Lofoten Basin, the present study investigates how submesoscale dynamics enhance vertical heat transport in this eddy-rich region. Submesoscale motions, typically with lateral scales of order 0.1–10 km (the upper bound corresponding to the local deformation radius) and timescales of a few inertial periods (~days), are less constrained by Earth's rotation than larger-scale motions and therefore often associated with ageostrophic secondary circulation, enhancing vertical transport[19,21]. In particular, the edges of mesoscale eddies are hotspots of submesoscale vertical transport, where the straining of mesoscale eddies can sharpen the lateral buoyancy gradients and drive ageostrophic secondary circulations – a submesoscale generation mechanism known as frontogenesis[21,22]. In eddy-rich regions such as the Kuroshio Extension, Gulf Stream, and Antarctic Circumpolar Current, submesoscale motions are found to be important contributors to vertical velocities (~10–100 m d$^{-1}$) in the upper ocean[20,23,24]. They provide a pathway for vertical exchange of heat, nutrients and organisms between the surface and deep oceans[19,20]. In the Lofoten Basin, which is also eddy-rich, a recent study[25] has focused on vertical transport mechanisms driven by mesoscale eddies, e.g., eddy pumping (1 m d$^{-1}$) and eddy-induced Ekman pumping (0.1 m d$^{-1}$), while the role of submesoscale vertical transport remains poorly documented and quantified due to the lack of extensive and long-term fine-scale in situ observations in this region.

We address this observational challenge by utilizing a fine-scale, Seaglider dataset over five years, which repeatedly traversed the mesoscale eddies in the Lofoten Basin. The motivation for focusing on the VHT associated with submesoscale motions is the frequent satellite-based detection of positive sea surface temperature anomalies (SST$_a$) along the edges of mesoscale eddies. Intriguingly, positive SST$_a$, forming "warm ring" structures of 1–10 km scale, were predominantly observed at the edges of both cyclones and anticyclones, with negative SST$_a$ only rarely occurring. The combination of altimeter-based eddy dataset with Seaglider observations enables an

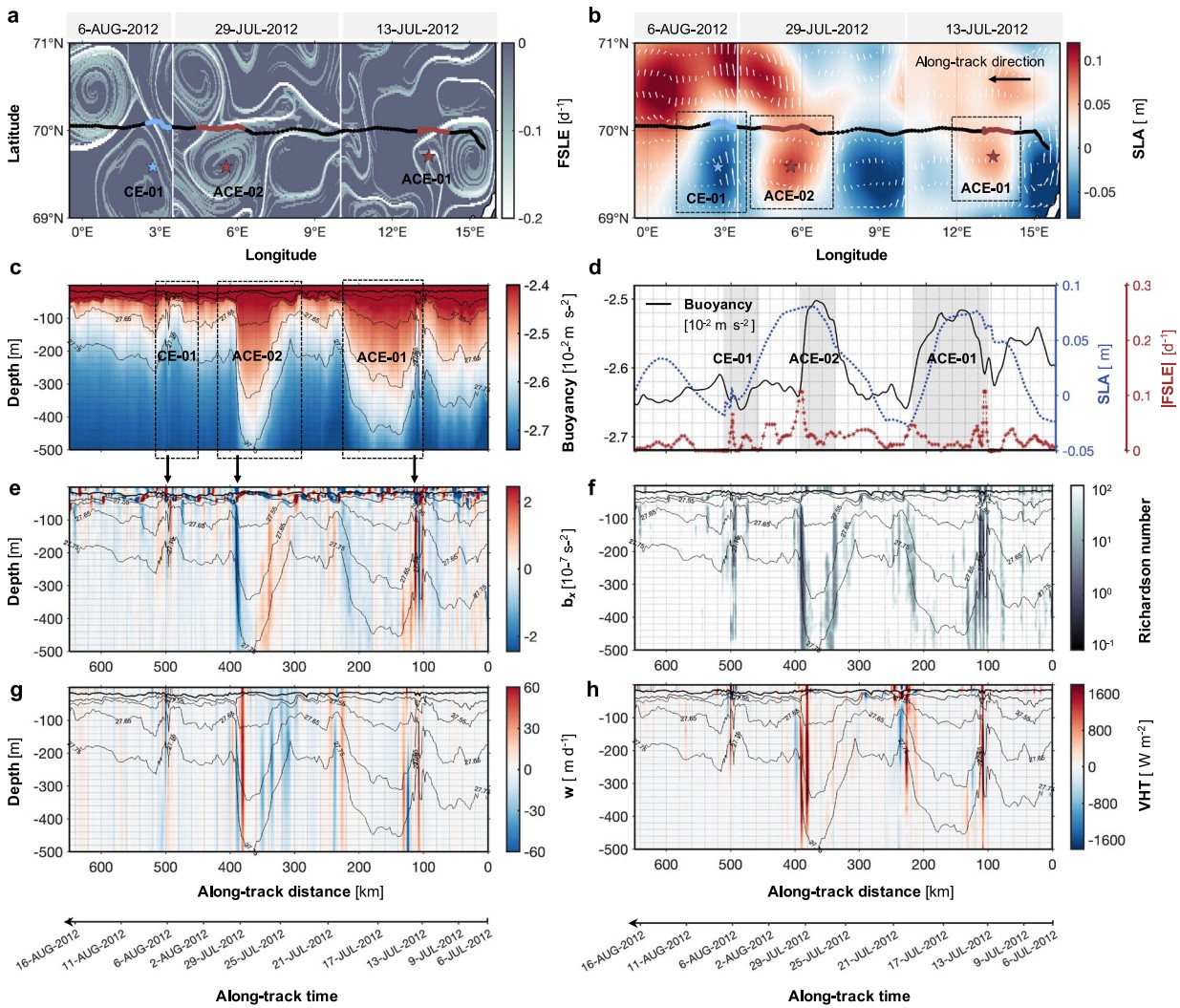

**Fig. 2 | Characteristics of submesoscale ageostrophic motions at the edges of eddies. a** Finite-size Lyapunov exponent (FSLE) and **b** Sea level anomaly (SLA) superimposed on the Seaglider track from 6 July 2012 to 16 August 2012. ACE-01, ACE-02, and CE-01 represent the three eddies traversed by the glider on 13 July 2012, 29 July 2012 and 6 August 2012, respectively. The FSLE and SLA fields also show snapshots of these three days. The glider's vertical section of **c** buoyancy,

**e** lateral buoyancy gradient ($b_x$), **f** geostrophic Richardson number ($Ri$) and **g** vertical velocity ($w$). **h** The glider's vertical section of vertical heat transport (VHT), with positive (negative) values denoting upward (downward) heat transport. The mixed layer depth (MLD) is shown by a thick black line in (**c**, **e**–**h**). **d** An along-glider-track time series of averaged buoyancy (black line), SLA (blue line) and |FSLE| (red line).

exploration of whether these "warm rings" are driven by submesoscale ageostrophic motions occurring below the sea surface, and the role of geostrophic strain in the meso-submesoscale interactions. Building on this intriguing observation, a key contribution of our study is the isolation and comparison of VHT in both cyclones and anticyclones. We investigate how deep-reaching submesoscale motions drive a systematic upward heat flux along their respective edges, providing a dynamical explanation for the "warm ring" phenomenon in a comparative context that has been largely overlooked.

## Results

### Submesoscale VHT revealed by Seaglider observations

Between July 2012 and August 2017, eight high-resolution Seaglider missions (M1–M8) amassed over 44,535 km of temperature and salinity data along profiles in the eddy-abundant regions of the LB (Fig. 1, Supplementary Tables S1, and Supplementary Fig. S1). A cross-basin example is the M1 mission conducted from 6 July 2012 to 16 August 2012 Fig. 2). The sg559 glider was deployed near the continental slope and traversed over 600 km over one month on a nearly straight trajectory towards the west, recording data continuously from the

continental slope to the center of the LB (70°N, 0-16°E). The glider passed numerous mesoscale eddies during the mission, and particularly it traversed two anticyclonic eddies and one cyclonic eddy on 13 July, 29 July, and 16 August, respectively (Fig. 2a–c and Supplementary Fig. S2). Frontal structures, characterized by large lateral buoyancy gradients ($|b_x| > 2 \times 10^{-7}$ s$^{-2}$), were identified at the edges of these eddies along the glider track (approximately at 100, 400 and 500 km in Fig. 2e). These structures exhibit a strong spatial correspondence with the intense stretching identified in the altimetry-based finite-size Lyapunov exponent (FSLE) (Fig. 2a, d, e). FSLEs indicate the timescale of the stretching and compression induced by the strain field[10,26,27] and, as such, strong FSLE are expected to be co-located with strong lateral gradients of buoyancy. This is indeed the case, as strong stretching FSLE characterized by large negative values (<−0.15 d$^{-1}$), is distributed along the eddy edges, while the values inside and outside the eddy are much weaker, close to zero at the eddy center. The width of the frontal structure (Fig. 2e) ranges from 2 to 10 km, falling within the submesoscale range. To investigate the dynamical mechanism driving the formation of strong lateral buoyancy gradients, we compute the geostrophic Richardson number[23,28] ($Ri = f^2 N^2 / b_x^2$). The small $Ri$ values

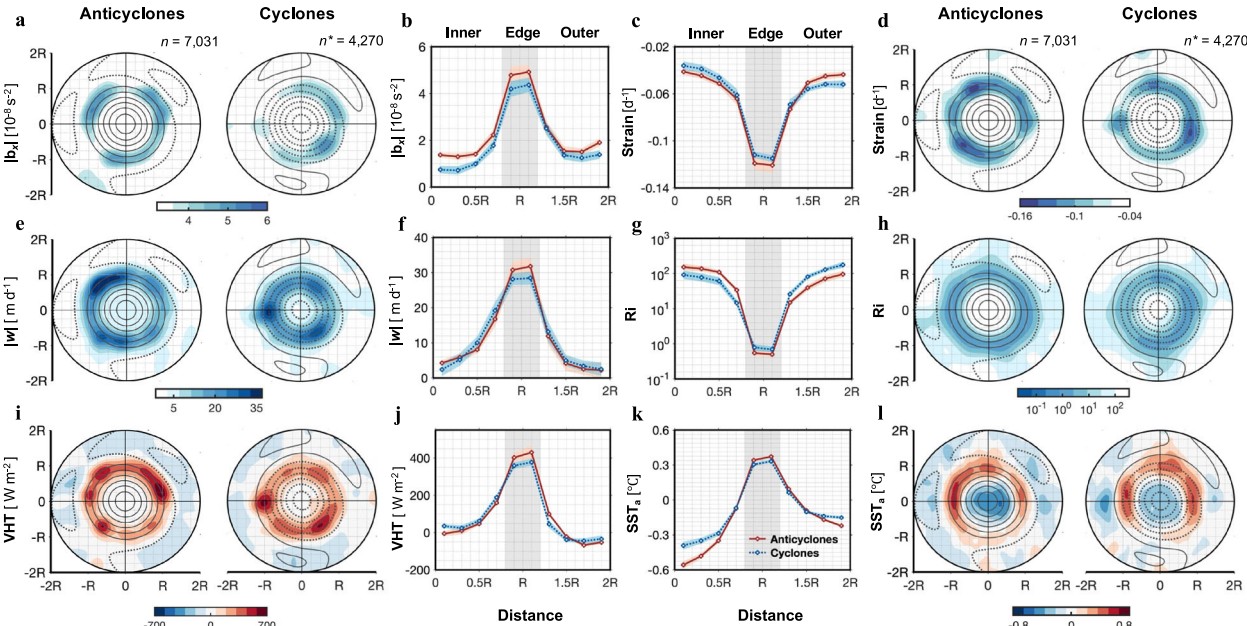

**Fig. 3 | Upward vertical heat transport (VHT) and the "warm wing" associated with submesoscale motions.** In normalized coordinates, panels **a** and **b** show the lateral buoyancy gradient ($|b_x|$), **c**, **d** strain, **e**, **f** vertical velocity ($|w|$), **g**, **h** geostrophic Richardson number ($Ri$), **i**, **j** VHT, and **k**, **l** satellite-based sea surface temperature anomalies ($SST_a$). Panels **a**, **d**, **e**, **h**, **i**, and **l** correspond to eddy-centric composite distributions (0–500 m averaged), while panels **b**, **c**, **f**, **g**, **j**, and **k** represent radial distributions from the eddy center to 2R, with the colored shaded areas indicating the standard error. $R$ represents the normalized radius of the eddy. $n$ and $n^*$ represent the number of Seaglider sampling profiles associated with anticyclonic and cyclonic eddies, respectively. Positive (negative) sea level anomaly (SLA) is represented by solid (dashed) black contours. Details on the projection and collocation processes are provided in Supplementary Fig. S7.

($Ri \leq 4$, Fig. 2f) observed at the same locations indicate that energetic ageostrophic motions associated with intense vertical currents are occurring at the edges of eddies (see Methods for a full description).

The vertical section of variations in vertical velocity, '$w$', can reach 60 m d$^{-1}$ (Fig. 2g), indicating strong vertical transport. The $w$ field inherits the same intermittency observed in FSLE and $b_x$ fields (Fig. 2a, e, g), with high magnitudes primarily found at the edges of the eddies. These strong vertical currents extend their influence well into the ocean interior−reaching depths of up to 500 m, which far exceeds the mixed layer (~20 m deep in summer). VHTs were further estimated by using the submesoscale components of vertical velocity and temperature (Fig. 2h). Although VHTs associated with mesoscale eddies are intensively distributed within the mixed layer, below the mixed layer the VHTs dominate at the locations of submesoscale fronts, consistent with a previous study[23]. Their local amplitude reaches 1600 W m$^{-2}$ below the surface mixed layer and extends to at least 500 m. The intense VHTs located at the edges of the eddies are predominantly positive, indicating that these motions can drive heat transport from the ocean interior upward to the surface (see other transect cases in Supplementary Figs. S3, 4). These analyses of intense $w$ and VHT are further supported by regional numerical simulations with a 2.4 km horizontal resolution (see details in Supplementary Notes). Model results from two representative cases in the same region similarly reveal strong vertical flows along both anticyclonic and cyclonic eddy edges, extending to depths exceeding 600 m (Supplementary Figs. S5, 6). These vertical velocities exhibit distinct paired structures of upward and downward flows, consistent with the observed patterns and indicative of strong secondary circulation at the eddy edges that facilitates heat transport from the ocean interior toward the surface.

## Submesoscale VHT forms "warm rings" in both cyclones and anticyclones

While submesoscale signals can be obtained through Seaglider observations of individual eddies, the consistent features of the submesoscale emerge when compositing observations from a large number of section traverses through eddies Fig. 3 and Supplementary Fig. S7. Figure 3a−d shows the composite distribution of absolute lateral buoyancy gradients ($|b_x|$) and stretching induced by geostrophic strain in the normalized eddy-centered coordinates. An obvious feature is that the strong $|b_x|$ band is located within a ring-shaped area around the eddy with larger magnitudes in the anticyclonic eddy than the cyclonic case (Fig. 3a, b). The horizontal distribution of the strain field is almost identical to $|b_x|$ (Fig. 3a−d). This suggests the role of the strain in sharpening fronts[27,29], i.e., stretching the fluid along the front (FSLE ridges) while compressing the fluid in the direction across the front. This front-sharpening mechanism explains why the gliders captured strong lateral buoyancy gradients when traversing these edges (fronts) Figs. 2e, 3a, b and 4a, b). As the cross-front scale is compressed by the strain field, frontogenesis intensifies, increasing the buoyancy gradient and ultimately disrupting the thermal wind balance[19,30]. To restore this balance, an ageostrophic secondary circulation develops, characterized by surface convergence with downwelling on the dense side and upwelling on the lighter side of the front (Fig. 2g and Supplementary Figs. S3–6). This secondary circulation generates vertical velocities reaching up to 60 m day$^{-1}$ below the surface mixed layer, nearly an order of magnitude greater than those associated with mesoscale motions (Fig. 2g, Supplementary Fig. S8, and Supplementary Notes).

A three-dimensional view of buoyancy anomalies measured by Seagliders reveals distinct vertical structures characteristic of mesoscale eddies at depth (Fig. 4). Anticyclonic eddies exhibit positive buoyancy anomalies forming a bowl-shaped structure, whereas cyclonic eddies display negative anomalies forming a doming structure. These patterns reflect different dynamics: anticyclonic eddies (high SSH) trap warm, saline waters driven by surface convergence and downwelling, whereas cyclonic eddies (low SSH) contain colder, fresher waters resulting from surface divergence and upwelling (Fig. 2c and Supplementary Fig. S2). The vertical distributions of temperature and salinity (Supplementary Figs. S1, 2) reveal the region's typical

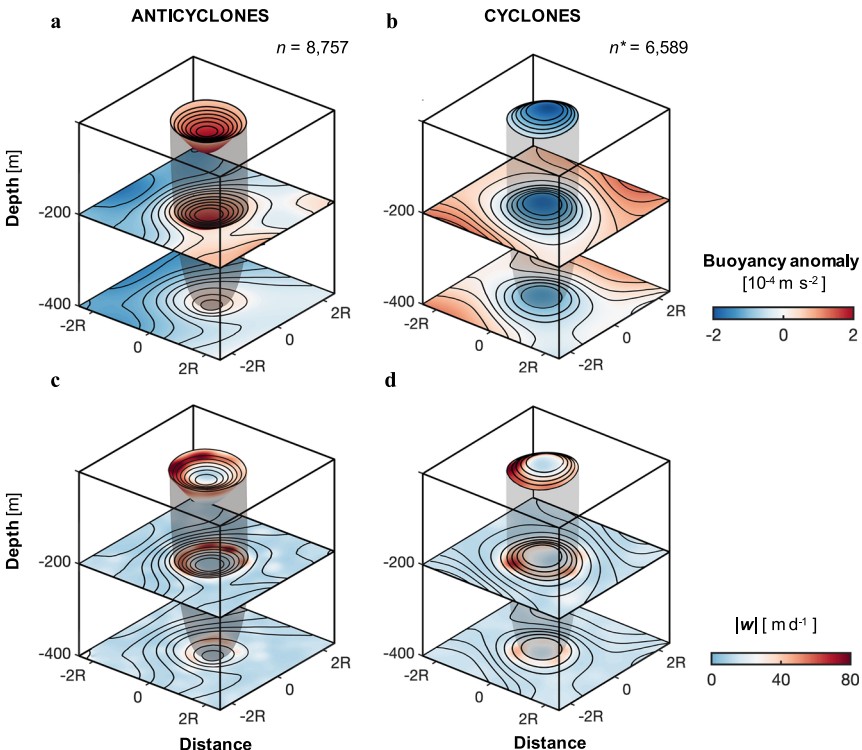

**Fig. 4 | 3D structures of buoyancy anomalies and vertical velocities for anticyclonic and cyclonic eddies.** Eddy-centric composite distributions of buoyancy anomalies at 0 m, 200 m and 400 m depth for **a** anticyclonic and **b** cyclonic eddies based on five years of Seaglider observations. Positive (negative) buoyancy anomalies within **a** anticyclonic (**b** cyclonic) eddies defined by altimetry data represent warm-core (cold-core) structures. The transparent black shading represents the outermost layer of the eddy, defined by closed contours of surface buoyancy anomalies. **c** and **d** are same as **a** and **b** except for eddy-centric composite distributions for vertical velocities. The contours in (**a**, **c**) and (**b**, **d**) represent the buoyancy anomalies (contour interval $2 \times 10^{-5}$ m s$^{-2}$) for anticyclonic and cyclonic eddies, respectively.

hydrographic structure, with temperature variations ($\Delta T \approx 2$–4 °C) substantially exceeding salinity variations ($\Delta S < 0.1$–0.2). The dominant role of temperature in controlling density is quantified by the density ratio, $R_\rho = (\alpha \Delta T)/(\beta \Delta S)$. Using representative values for thermal expansion ($\alpha \approx 2 \times 10^{-4}$ °C$^{-1}$) and haline contraction ($\beta \approx 8 \times 10^{-4}$ psu$^{-1}$) pertinent to these waters, $R_\rho$ is approximately 5 for these observed variations. This high ratio indicates that, despite the high-latitude (69–72°N) Atlantic Water setting, temperature predominantly governs density structure below the mixed layer down to ~500 m depth. Complementary analyses of vertical velocity further highlight intense vertical motions predominantly driven by submesoscale processes along the edges of both anticyclonic and cyclonic eddies, extending downward to at least 400 m (Fig. 4c, d, Supplementary Fig. S8, and Supplementary Notes). The low geostrophic Richardson number ($Ri \leq 4$) are co-located with the strong geostrophic strain and the large magnitudes of $|w|$, further indicating an energetic ageostrophic regime characterized by intense vertical motions (Fig.3c–h). These results are consistent with the expectation that geostrophic strain shrinks the frontal scale and intensifies lateral buoyancy gradients, disrupting thermal wind balance and thereby promoting cross-frontal secondary circulations in the frontal zone.

We explored the impact of deep-reaching submesoscale vertical currents on the domain-averaged VHT by constructing eddy-centered VHT composite fields. The composites (Fig. 3i, j and Supplementary Fig. S9) reveal upward VHT at the eddy edges where the geostrophic strain is enhanced, transporting heat from ocean interior to the surface. Remarkably, the magnitudes of the upward VHT in both cyclonic and anticyclonic eddies are comparable, reaching an average of ~400 W m$^{-2}$ (Fig.3i, j). This occurs because, irrespective of being cyclonic or anticyclonic, the submesoscale secondary circulations act to flatten sloping isopycnals at the fronts[20,30], thus restoring stratification[31]. These secondary circulations are characterized by upwelling on the warmer (lighter) side and downwelling on the colder (denser) side of the front, resulting in consistently upward heat transport ($w' T' > 0$), as further explained in the next section. A two-year composite analysis from numerical simulations also demonstrates the structure, depth range, and intensity of $w$ and VHT, which are consistent with the observational results, further supporting our findings (Supplementary Fig. S10). To further examine whether the intense upward heat transport at the deep-reaching fronts affects the SST, we constructed composite averages of submesoscale SST$_a$. Prior to the composite analysis, SST$_a$ fields were high-pass filtered spatially to remove large-scale circulation features. Results indicate that these deep-reaching submesoscale fronts indeed lead to an additional increase of ~0.4 °C in SST$_a$ along the edges of eddies (Fig. 3k, l).

**Geostrophic strain-induced frontogenesis drives an intense ageostrophic secondary circulation and upward VHT**

To identify the role of geostrophic strain in enhancing submesoscale VHT, the composite result of strain was organized into three categories based on strain intensity (Group 1: $S_n \in [P_{40}, P_{60}]$, Group 2: $S_n \in [P_{20}, P_{40}]$, Group 3: $S_n \in [P_0, P_{20}]$) along the edge of eddies Fig. 5a). The composite results for VHT and SST$_a$ corresponding to these three groups of strain intensity are presented in Fig. 5b, c. As the intensity of the strain increases, the strength of the VHT also rises. Particularly, the group with the strongest strain (Group 3) is associated with the strongest upward heat transport in a ring-shaped region. This is attributed to frontogenesis, whereby strong geostrophic strain intensifies lateral buoyancy gradients along the eddy edges. An ageostrophic secondary circulation develops in response, acting to flatten these gradients and thereby enhances systematic upward heat transport. Further analyses of satellite SST$_a$ composites (Fig. 5c) indicate

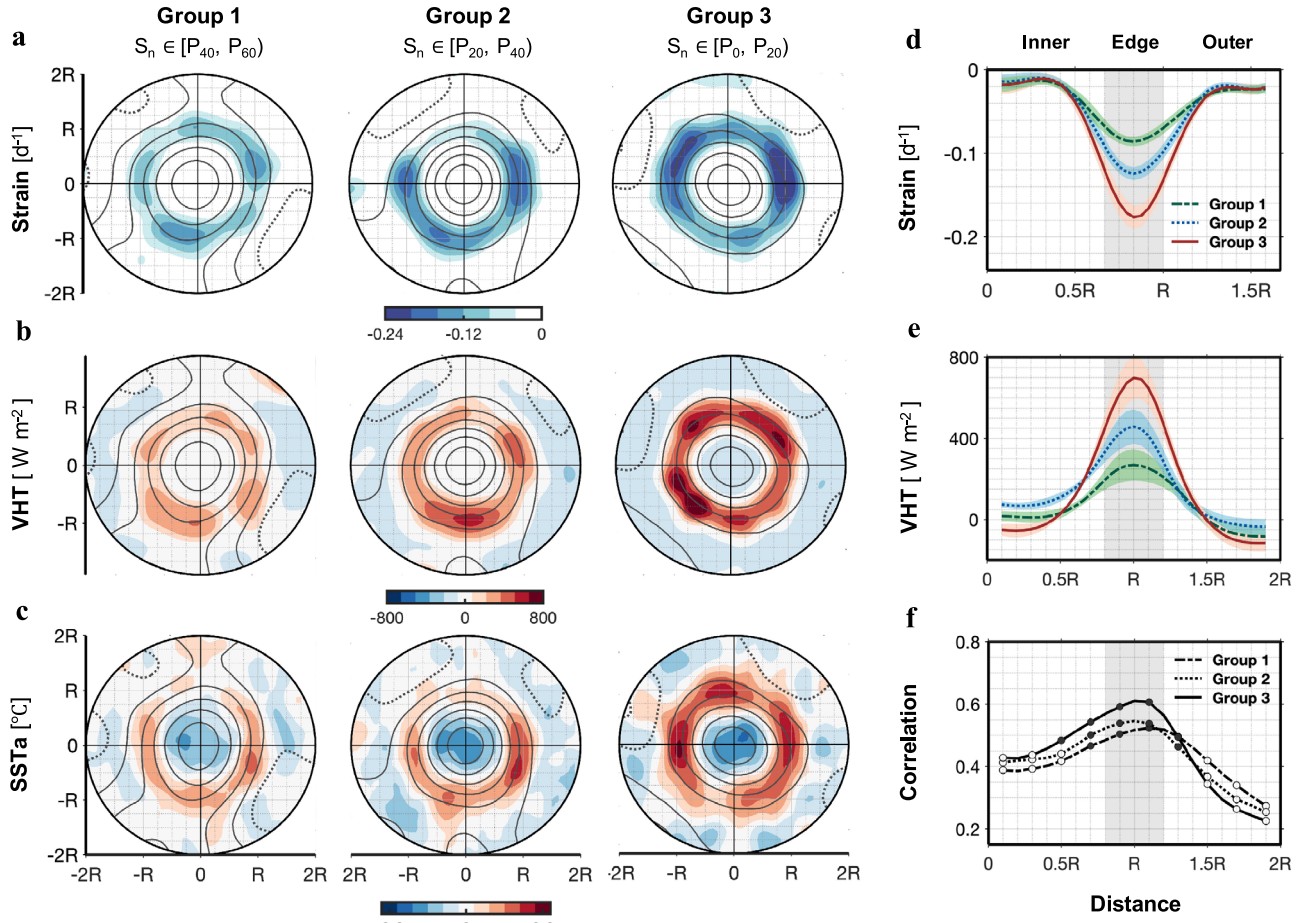

**Fig. 5 | Geostrophic strains affect the intensity of submesoscale motions and vertical heat transport (VHT) along the edge of eddies. a** Eddy-centric composite distributions averaged over 0–500 m for different strain intensities ($S_n$): Group 1 ($S_n \in [P_{40}, P_{60})$), Group 2 ($S_n \in [P_{20}, P_{40})$), and Group 3 ($S_n \in [P_0, P_{20})$), where $P_0$, $P_{20}$, $P_{40}$, and $P_{60}$ represent the 0th, 20th, 40th, and 60th percentiles of the strain distribution, respectively. Corresponding eddy-centric composite distributions of **b** VHT and **c** sea surface temperature anomalies ($SST_a$) for different strain intensity groups. Radial distribution of the magnitude of **d** Strain, and **e** VHT for different strain intensity groups from eddy center to $2R$. **f** Correlations between strain and VHT for different strain intensity groups from eddy center to $2R$. Correlations significant at the 95% confidence level are plotted as solid dots, and the others are plotted as open dots. $R$ represents the normalized radius of the eddy.

that $SST_a$ due to submesoscale motions are correlated with geostrophic strain, confirming that geostrophic strain is a crucial mechanism that facilitates the transfer of heat from the ocean interior to the surface.

Figure 6 illustrates how the development of submesoscale secondary circulation and upward VHT is driven by strain fields within hyperbolic regions. Specifically, these hyperbolic regions are located at the edges of mesoscale eddies, near areas of maximum azimuthal velocities, and can be identified by Lagrangian Coherent Structure (LCS)[25,26,32]. The most effective way to enhance a geostrophic front and trigger ageostrophic perturbations is by stretching along the front direction and compressing across the front direction[30,31]. As depicted in Fig. 6a, the strain field (black arrows) from $t = 0$ to $t = 3$ elongates the tracer patch in the $y$-direction (along the front) and compresses it in the $x$-direction (across the front). This process intensifies the buoyancy gradient while simultaneously reducing the tracer patch's scales along the $x$-direction. The time-dependent LCS theory[26] explains the fluid stretching mechanisms critical for forming submesoscale fronts. As strain fields compress the front, the cross-frontal buoyancy gradient intensifies and the cross-frontal scale decreases, driving frontogenesis and increasing the available potential energy at the front. When the local Rossby number ($R_o = \zeta/f$) approaches $O(1)$, thermal wind balance becomes disrupted, triggering ageostrophic secondary circulations that act to restore this balance. These circulations generate upwelling

on the warm (light) side and downwelling on the cold (heavy) side of the front. This is accompanied by a cross-frontal surface flow moving from the warm to the cold side, and a cross-frontal deep flow moving from the cold to the warm side. The role of this cross-frontal ageostrophic secondary circulation (orange arrows in Fig. 6b) is to flatten the isopycnals and reduce the buoyancy gradient. Typically, warm, less dense waters occupy anticyclonic eddy interiors and cyclonic eddy exteriors, while cold, dense waters characterize cyclonic eddy interiors and anticyclonic eddy exteriors. As a consequence, the enhanced VHT associated with frontogenesis at the edges of both anticyclonic and cyclonic eddies is directed upward due to the positive correlation ($w'T' > 0$) between temperature and vertical velocity anomalies (Fig. 6b). The combination of satellite-based FSLE with glider-based vertical measurements provides observational evidence for diagnosing how eddy strain fields drive strong buoyancy gradients and vertical transport at submesoscales in the ocean interior. The continuous upward VHT within strain-dominated regions transports heat from the ocean interior back to the surface, thereby elevating sea surface temperatures (SST) along the edges of eddies.

## Discussion

Our results emphasize the crucial role of submesoscale motions within mesoscale eddies in enhancing VHT in an eddy-rich ocean such as LB, a role that has been long neglected. Mesoscale eddies, along with the

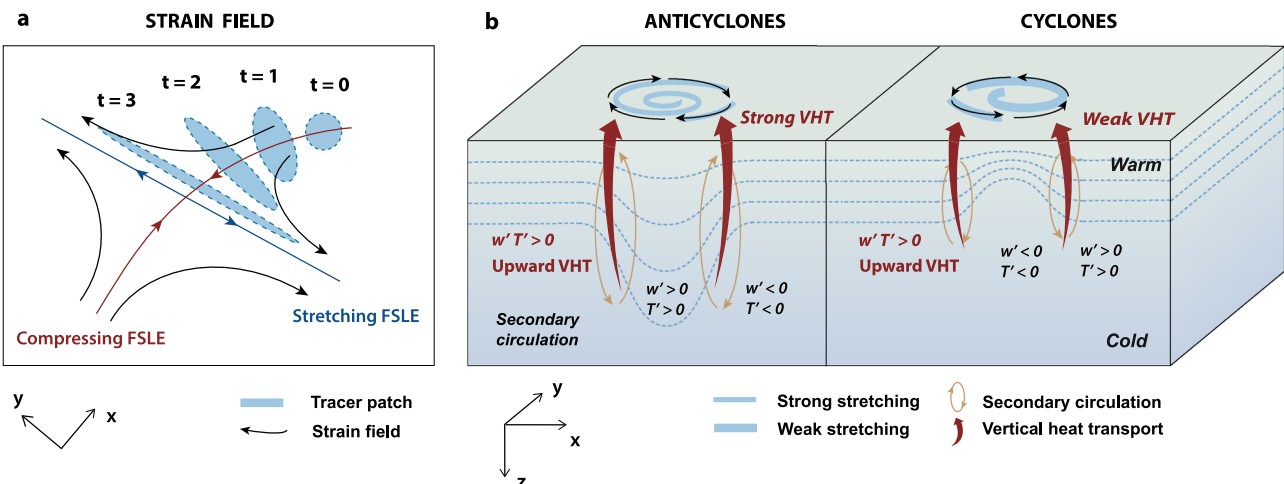

**Fig. 6 | Strain-induced frontogenesis, secondary circulations, and upward vertical heat transport (VHT). a** Time-dependent horizontal field of deformation of a tracer patch (blue patch) in a strain field (black arrows). Finite-size Lyapunov exponent (FSLE) arrows represent horizontal stretching (blue, $y$-direction) and compression (red, $x$-direction). The tracer patch stretches along the $y$-direction due to the strain from $t = 0$ to $t = 3$. **b** 3-D conceptual diagrams of strain-induced frontogenesis processes and submesoscale VHT along the edges of anticyclonic and cyclonic eddies. The submesoscale fronts caused by mesoscale geostrophic strain (aligned with the stretching FSLE). Vertical velocities ($w$, orange arrows) develop with time and frontogenesis ($x$-$z$ plane). The blue dashed lines are isotherms. As the temperature and $w$ anomalies are positively correlated ($w'T' > 0$), frontogenesis-induced VHT is upward at the edges of both anticyclonic and cyclonic eddies, with its magnitude (strong or weak) depending on the local strain rate.

mean flow, have traditionally been considered primary contributors to oceanic heat transport through lateral processes, such as advection, stirring and mixing[33–36]. Positive temperature anomalies associated with subsurface anticyclonic eddies have also been observed at the surface in Mediterranean water eddies (Meddies)[37] and the LBE[38]. However, while these lateral transport processes remain important, previous work has identified enhanced submesoscale heat fluxes in strain-dominated regions[23]. Building upon this, our study reveals that within mesoscale eddies, vertical transport is largely governed by submesoscale ageostrophic motions along both cyclonic and anticyclonic edges (Supplementary Figs. S8, 9 and Supplementary Notes). Specifically, the continuous vertical structures of $w$ are primarily observed in the strain-dominated areas around eddies, serving as a crucial pathway for the vertical exchange of heat and energy between the ocean interior and the surface.

Previous studies have identified several vertical transport mechanisms associated with mesoscale eddies[18,35,36], including eddy pumping and eddy-induced Ekman pumping, which generate vertical velocities of $O(1\,\mathrm{m\,d^{-1}})$ and $O(0.1\,\mathrm{m\,d^{-1}})$ respectively within eddy cores. Specifically, in the Lofoten Basin, numerical and observational studies have emphasized the importance of vertical motions in the dynamics of the long-lived LBE[1,25,38]. A model with 4-km horizontal resolution revealed that centrifugal force-driven divergence drives upwelling within the upper LBE, compensated by downwelling in peripheral regions- a mechanism linked to high strain at the LBE peripheries intensifying toward the sea surface[38]. Bosse et al.[1] revealed that potential vorticity (PV) in the LBE core is reduced by two orders of magnitude relative to surroundings, with high strain and strong PV gradients at peripheries forming a dynamical barrier[1]. They suggest that these peripheries are likely accompanied by intense submesoscale dynamic processes, which aligns with our findings.

While autonomous gliders provide extensive spatial and temporal coverage, their relatively slow speed compared to submesoscale phenomena can introduce Doppler smearing effect, particularly from internal wave contamination[39]. We implemented objective interpolation (OI) and spatial filtering to mitigate these effects, and validated the effectiveness of this processing approach with an independent along-isopycnal analysis that yields consistent dynamical fields (Supplementary Figs. S11e, f, 12 and Supplementary Notes). Using Seaglider

observations and high-resolution simulations, we compared submesoscale and mesoscale components (Supplementary Figs. S8–10 and Supplementary Notes) and found that submesoscale vertical velocities (~60 m d⁻¹) substantially exceed mesoscale vertical velocities (<10 m d⁻¹) by nearly an order of magnitude, consistent with previous reports[18,20,22]. Similarly, submesoscale VHT (up to 1400 W m⁻²) is considerably stronger than mesoscale transport (Supplementary Fig. S8). Although mesoscale processes play an important role in vertical heat exchange, submesoscale motions generate net VHT exceeding mesoscale contributions by more than an order of magnitude. Even without considering net heat flux, mesoscale VHT is only 1/6 to 1/5 of the submesoscale contribution (Supplementary Notes). The combined effect of these processes within the eddy region results in a net upward heat transport[40–42]. Spatially, submesoscale processes are predominantly concentrated at eddy edges, creating the observed "warm-ring" structures, while mesoscale processes are more broadly distributed across both eddy edges and interiors, typically forming classical "warm-core" or "cold-core" structures (Supplementary Figs. S9, 10 and Supplementary Notes). Remote sensing chlorophyll results support these pumping processes: low chlorophyll concentrations within cyclones and high chlorophyll concentrations within anticyclones (Fig. 1a), while at eddy edges, elevated chlorophyll concentrations are often found at cyclone edges and decreased concentrations at anticyclone edges, with positive $SST_a$ at the edges of both. These patterns suggest[19,29] that material and heat transport at eddy edges is predominantly governed by vertical rather than horizontal processes such as mesoscale eddy stirring.

Despite challenges in measuring vertical velocities in ocean currents, numerous studies[19,20,23] have employed the Omega equation and numerical simulation to estimate these velocities in various regions. A study[40] employing a fine-resolution global model demonstrates that submesoscale heat transport can increase sea surface temperature by up to 0.3 °C in the mid-latitudes. Siegelman et al.[23] combined mammal observations with Lyapunov exponents to diagnose enhanced upward heat transport at deep submesoscale fronts in the Antarctic Circumpolar Current region, finding similar results as our study. Our contribution, however, further incorporates the eddy composite method to elucidate the dynamical processes of VHT driven by the frontogenesis of mesoscale eddies with opposite polarities, and

provides a 3D view of ocean dynamics from meso- to submesoscales. Numerical simulations strengthen these findings, showing that the vertical structure and depth of $w$ and VHT closely resemble observational results, with positive VHT predominating at eddy edges (Supplementary Figs. S5, 6, 10). Frontogenesis-induced continuous upward heat transport can lead to a 0.4 °C increase in SST along the edges of both cyclonic and anticyclonic eddies. In the Lofoten Basin, the most intense air-sea interaction zone in the Nordic Seas with active eddy dynamics, this submesoscale-dominated upward heat transport within the ocean interior can exceed three times the magnitude of local air-sea heat flux[3,43,44]. These results indicate that the long-neglected submesoscale VHT in the upper ocean are likely a key reason for the substantial heat loss in Atlantic Water in eddy-rich regions.

## Methods
### Satellite data
Daily Sea Level Anomaly (SLA) and geostrophic current velocity ($u_g$, $v_g$) product with a $0.25 \times 0.25°$ resolution were utilized to identify mesoscale eddies in the Lofoten Basin area (0–16°E, 69–72.2°N). The Level-4 gridded altimetry products from Copernicus Marine Environment Monitoring Services (CMEMS, http://marine.copernicus.eu) were constructed by merging data from missions including TOPEX/Poseidon, Jason-1/2, ERS-1/2, GFO, CryoSat-2, HY-2A, Altika and ENVISAT. At our study latitude (~70°N), this gridded product yields a meridional resolution of ~27.8 km and a zonal resolution of ~9.5 km. For the mesoscale eddies with radii of 30–50 km that we focus on, the zonal resolution is sufficient to resolve their horizontal velocity structure given their quasi-circular nature. Our previous study[45] quantitatively evaluated this dataset and found that altimeter-derived geostrophic velocities within eddies were ~25–30% lower than the geostrophic velocities concurrently measured by surface drifters within the eddies. This discrepancy was mainly attributed to the smoothing effect inherent in the gridding of altimeter data, along with additional influences on the drifter velocities from centrifugal acceleration and wind-slip effects.

Level-2 SST products from VIIRS Suomi-NPP and MODIS Aqua were taken from the NASA Ocean Color achive (https://oceancolor.gsfc.nasa.gov). The products from VIIRS and MODIS, with the spatial resolutions of 750 m and 1 km, respectively, were used to construct $SST_a$ associated with the mesoscale eddies. Satellite altimetry and SST-based products were selected to be consistent with the data collected during the Seaglider operations from July 2012 through August 2017.

### Stretching finite-size Lyapunov exponent (FSLE)
FSLEs, derived from satellite-based geostrophic velocities, were employed to characterize the properties of the strain field. The FSLEs are computed from the time interval $\tau$, during which two fluid particles move from an initial separation distance $\delta_i$ to a final separation distance $\delta_f$ following their trajectories in the two-dimensional velocity field. At time t and position $x$, the Lyapunov exponent $\lambda$ is defined as[26,32,46]:

$$\lambda\left(x, t, \delta_i, \delta_f\right) = \frac{1}{\tau} \log\left(\frac{\delta_f}{\delta_i}\right) \quad (1)$$

where the position vector $x$ is the independent variable. The Lyapunov exponent $\lambda$ quantifies the local exponential rate of separation between initially nearby trajectories starting at position $x$ over the integration time $\tau$. The separation distances ($\delta_i$, $\delta_f$) are expressed in degrees as angular distances on the Earth's surface, and the ratio $\delta_f/\delta_i$ becomes dimensionless after taking the logarithm, thus the units of FSLEs are $d^{-1}$. Trajectories were extracted by applying a fourth-order Runge-Kutta scheme with a time step of 3 h. Values of $\delta_i$ and $\delta_f$ were set at 0.02° and 0.4°, respectively, to capture meso- to submesoscale properties and to visualize structural details[10,25].

We computed the daily FSLE field over the five-year period of Seaglider operations to quantify the stretching induced by geostrophic strain in the regions traversed by Seaglider. By combining these data with the altimetry-based eddy dataset, the strain characteristics at the edges and within the eddies were further extracted and analyzed. Negative (positive) FSLE values indicate that a tracer patch is stretched (compressed) in the $x$ ($y$) direction of the background strain field (Fig. 6a). Consequently, greater negative (positive) FSLE values correspond to more intense stretching (compression). Here, negative FSLE values were computed back in time as detailed in Dong et al.[10,25,46] and d'Ovidio et al.[26,47].

### Seaglider measurement
We analyzed in situ data collected between 4 July 2012 and 9 August 2017 through eight Seaglider missions (M1-8, see Fig. 1) in the Lofoten Basin[48–50]. The initial six glider deployments (M1-6) were part of the Norwegian Atlantic Current Observatory (NACO), dedicated to observing the variability of the North Atlantic Ocean Current[48]. The subsequent deployments (M7-8) were part of the ProVoLo project[49]. The Seaglider missions consisted of two main parts. The first part included 24 cross-basin transects (0–16°E) gliding in an east-west direction, primarily between the continental slope and LB, including some meandering paths. These transects covered a total length of 15,781 km, accounting for ~35% of the total distance and 45% of the total mission time. The second part involved repeated sampling on spiral and circular trajectories in the deepest areas of LB, covering 25,685 km, representing nearly 57% of the transects but only 44% of the time. These tracks covered the western side of the Norwegian Atlantic Slope Current and the LB region, where mesoscale eddy activity is most frequent and eddy intensity is strongest[11,12,45]. All Seaglider data have undergone quality control and post-deployment calibration (Supplementary Notes). Combined with satellite-derived eddy locations found that over the five-year time period, Seagliders collected vertical data on more than 1783 eddies, of which 760 were cyclones and 1023 were anticyclones. The glider collected data from 7031 sampling points associated with cyclonic eddies and 4270 sampling points associated with anticyclonic eddies (Supplementary Fig. S7). Over a total of 1543 days, these glider missions provided decent temporospatial coverage and resolution for studying the meso-submesoscale interactions in the LB.

### Mixed layer depth
The MLD was defined as the level of density increase of 0.03 kg m$^{-3}$ from the 15 m depth[50,51].

### Lateral buoyancy gradient
The along-track time series for buoyancy, defined as $b = -g(\rho - \rho_0)/\rho_0$, where $\rho$ is the potential density and $\rho_0 = 1025$ kg m$^{-3}$ as the reference density, demonstrates a variability encompassing both meso- and submesoscale processes. For further calculating the lateral buoyancy gradients, buoyancy ($b$) was first interpolated along the Seaglider's paths onto a regular grid with a horizontal resolution of 2 km and a vertical resolution of 5 m. The front was identified by the buoyancy gradient along the Seaglider direction, defined as $b_s = \partial b/\partial s$, where $s$ is the along-track distance. Considering that the Seaglider's trajectory typically inclines relative to the stretched FSLE streamlines, forming a certain angle with FSLE (streamlines rather than being perpendicular), and that the buoyancy front is assumed to align with the stretched FSLE, it became necessary to correct the Seaglider's buoyancy gradients to accommodate the direction of the FSLE encountered[23,52]. To achieve this, it is assumed that $\theta$ represents the angle between a FSLE eigenvector and a Seaglider's trajectory, and the normalized along-track lateral gradient of buoyancy ($b_x$) was calculated as $b_x = b_s / \sin\theta$. To focus on the regions prone to submesoscale motions, we excluded data with $\theta < 30°$, which

corresponds to a correction of more than a factor of 2, and only normalized $b_s$ associated with the large FSLE (>0.05 d$^{-1}$).

## Richardson number

To quantify the occurrence of the energetic ageostrophic motions associated with intense vertical velocities, we estimated the non-dimensional geostrophic Richardson number ($Ri$) using Seaglider observation data[19,23,28]. The geostrophic Richardson number is defined as $Ri = f^2 N^2 / b_x^2$, where $N^2 = \partial b / \partial z$ represents the Brunt-Väissälä frequency, and $b_x = b_s / \sin\theta = \partial b / (\partial s \sin\theta) = \partial b / \partial x$ represents the normalized along-track lateral gradient of buoyancy. Here, $Ri$ quantifies the relative importance of stratification versus the lateral buoyancy gradient, effectively characterizing the dynamic regime by comparing the steepness of isopycnal slopes to $N/f$. When $Ri \gg 1$, stratification dominates and the flow remains close to quasi-geostrophic (QG) balance[53,54]; conversely, when $Ri$ is on the order of unity (or $Ri \leq 4$), the weaker stratification relative to horizontal buoyancy gradient indicates an ageostrophic regime, characterized by intense vertical motions and active submesoscale dynamics[19,23].

## Vertical velocities and VHT

The ageostrophic vertical velocities ($w$) associated with mesoscale eddy fronts are diagnosed by numerically solving the adiabatic, two-dimensional (2D) version of the classical QG Omega equation (Supplementary Notes). The Omega equation was employed to solve for the gliding profile data of eight Seaglider missions within the 0–500 m depth range, aiming to obtain the variations of vertical velocities outside eddies, at eddy edges and within eddies. Buoyancy fronts are assumed to be elongated in the direction of stretching (Fig. 6), implying that the buoyancy gradient along the front ($y$-direction) is negligible compared to the gradient across the front ($x$-direction)[21,23,30]. Therefore, the 2D version ($x$, $z$) of the QG Omega equation can be expressed as[19,20,23,53]:

$$N^2 w_{xx} + f^2 w_{zz} = -2(u_x b_x)_x \qquad (2)$$

where the subscripts denote the derivatives. The variables $N^2$ and $b_x$ were computed using Seaglider data. The mesoscale strain field, as illustrated by the black arrows in Fig. 6a, stretches the tracer patch in the $y$-direction and compresses it in the $x$-direction. This leads to the development of intense horizontal buoyancy gradients at the submesoscale, with a growth rate being associated with the strain rate $u_x = \partial u / \partial x$. Therefore, we estimated the strain rate $u_x$ using the stretching FSLE derived from satellite altimetry data. Diverging from the traditional approach of directly estimating strain rate through $\partial u / \partial x$, FSLEs have the advantage of utilizing both the spatial and temporal variability of the velocity field derived from SLA[10,25]. As such, this Lagrangian diagnostic method provides information about both the growth rate and orientation of elongated horizontal buoyancy gradients by measuring the largest separation rate ($\lambda$) of two neighboring particles in the flow field. Here, we considered the background flow field to be barotropic. In Eq. (2), Seaglider trajectories were assumed to be perpendicular to the eddy fronts, achieved by normalizing the buoyancy gradient as described above. The vertical heat flux[23,40,41] is defined as:

$$\text{VHT} = \rho_0 C_p w' T' \qquad (3)$$

where $C_p$ is the specific heat capacity ($C_p$ = 3985 J kg$^{-1}$ K$^{-1}$), and $w'$ and $T'$ are the anomalies of vertical velocity and temperature, respectively, after removing the sectional mean.

## Identification of mesoscale eddies

To investigate the submesoscale motions at the edges of eddies in the Lofoten Basin, we constructed a mesoscale eddy dataset

spanning over three decades using satellite altimetry data[15,25]. The dataset was constructed with an automated hybrid eddy detection algorithm that combines geometric (Sea Level Anomaly, SLA) and dynamical (Okubo-Weiss, OW parameter) properties of the flow field to identify eddies[15,45,54–56]. The eddy center is defined as the location of the SLA maximum within the identified closed contour[45]. The eddy area corresponds to the region enclosed by the SLA contour associated with the maximum azimuthal (geostrophic) velocity, and the eddy radius ($R_x$) is defined as the radius of a circle with an area equivalent to this enclosed region. To accurately delineate eddy boundaries and areas, our hybrid detection method[15,25,45] employs the OW parameter ($W$) criterion to exclude spurious detections caused by noise or ambiguous, multipolar, or elongated contours, ensuring robust and consistent eddy boundaries. The OW parameter is defined as $W = (S_s^2 + S_n^2) - \varsigma^2$, where $S_s$ is the shearing deformation rate, $S_n$ is the stretching deformation rate, and $\varsigma$ represents relative vorticity. The total strain rate $S$ is calculated as $S^2 = (S_s^2 + S_n^2)$. Negative values of $W$ indicate dominance of relative vorticity over strain, a fundamental dynamical feature of mesoscale eddies. This hybrid detection algorithm significantly reduces reliance on arbitrary thresholds, thereby minimizing subjectivity and enhancing accuracy[15,25,45]. The method has been successfully applied and validated in the Lofoten Basin region using Argo floats and surface drifters[15,45,57]. Additional details are provided in Dong et al.[25] and Raj et al.[15,45].

## Eddy-centric collocations

To investigate the effects of submesoscale motions on VHT, we constructed eddy-centric composite fields[18,36,58] based on five years of Seaglider observations and the mesoscale eddy dataset, focusing on five parameters: $|b_x|$, strain field, $Ri$, $|w|$, and VHT. Composite fields were constructed using a normalized eddy-centric coordinate system, with $R$ representing a dimensionless normalized radius. $R_x$ represents the radius of an individual eddy. For each sampling point along the Seaglider transect, the radial distance ($d$) and azimuth angle ($\theta$) relative to due east were computed from latitude and longitude coordinates (Supplementary Fig. S7a). These sampling points were subsequently collocated onto the normalized eddy-centric coordinate system according to their normalized radial distance ($d/R_x$) and azimuth $\theta$ (Supplementary Fig. S7b), then interpolated onto a 100×100 grid for compositing. Our analyses primarily focused on the region within two radii of the eddy center ($d/R_x < 2$, Chelton et al.[54]), subdivided into three dynamical zones: eddy interior ($d/R_x < 0.8$), eddy edges ($0.8 \leq d/R_x \leq 1.2$), and eddy exterior ($d/R_x > 1.2$). This zone division was applied solely for descriptive and analytical clarity and was not employed in any data processing steps.

To provide a more intuitive demonstration of our composite analysis method, a case study figure is introduced to illustrate the single-eddy composite process (Supplementary Fig. S13). For this analysis, observational data from three specific eddies—ACE-02 and ACE-04 from the sg559-T2 transect, and ACE-01 from the sg561-T3 transect (Supplementary Figs. S3, 4)—were selected and then projected individually onto the normalized eddy-centric coordinate system after being averaged over the upper 500 m. The results clearly show that the strong signals of lateral buoyancy gradient ($b_x$) and vertical heat transport (VHT) originating from the eddy edges are consistently mapped to the annular region at a normalized radius of $r \approx R$. It is worth noting that our method does not perform any rotational alignment, thereby preserving the original azimuthal orientation of each feature.

To assess the impact of submesoscale VHT at the eddy edge on sea surface temperature, we constructed an eddy-centric composite analysis of SST$_a$ during the period of Seaglider operations. Daily SST maps were interpolated onto a 0.02° × 0.02° grid and high-pass

filtered with 0.2° latitude × 0.5° longitude half-power filter cutoffs to eliminate signals with length scales larger than the mesoscale variability[25,35,36]. This filtering process is denoted as $SST_{anom} = HP_{sp}(SST)$, where $HP_{sp}$ represents the high-pass filter. To reduce the influence of geographical locations and seasonal variabilities on the composite results, the long-term monthly averaged background fields at each longitude ($x$) and latitude ($y$) was subtracted from the $SST_{anom}$ at the corresponding location. The $SST_{anom}$ for the Seaglider transit were collocated to each identified eddy from the SLA for composite analysis. Eddies with over 70% coverage of available data within one radius of each identified eddy were included; those with less than 70% coverage were excluded. This minimizes the errors of the composites caused by cloud coverage within the eddy region[25]. Consistent with the coordinates of the Seaglider composites, the centers and radii of all eddies are normalized by the dimensionless radius $R$ to conduct composite analysis of $SST_a$ for eddies with different sizes and polarities.

## Data availability

All data used in this study are publicly available. Satellite altimetry data were obtained from the Copernicus Marine Environmental Monitoring Service (CMEMS, https://data.marine.copernicus.eu/product/SEALEVEL_GLO_PHY_L4_MY_008_047/services). Remote sensing SST and chlorophyll data were obtained from the NASA Ocean Biology Processing Group (Aqua MODIS mission, https://oceancolor.gsfc.nasa.gov/about/missions/aqua). Basemap land data for Fig. 1a were obtained from MODIS/Aqua via NASA Earthdata (https://doi.org/10.5067/MODIS/MYD09A1.061), and bathymetry from the ETOPO global relief model at 30-arc-second resolution (https://www.ncei.noaa.gov/products/etopo-global-relief-model). Seaglider observations are available via the Norwegian Marine Data Centre: Fer & Bosse (2017) (https://doi.org/10.21335/NMDC-UIB.2017-0001) and Bosse & Fer (2019) (https://doi.org/10.21335/NMDC-980686647).

## Code availability

The code used to produce the figures in this paper is available from the corresponding author upon request.

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

## Acknowledgements

H.D. was supported by the National Natural Science Foundation of China (NSFC, Grant No. 42306003), the Shanghai Science and Technology Innovation Action Plan (Rising-Star Sailing Program, Grant No. 23YF1418900), the China Postdoctoral Science Foundation (Grant Nos. 2023M732205 and 2023T160411), the Shanghai Postdoctoral Excellence Program (Grant No. 2022302), and the Shanghai Jiao Tong University Overseas Joint Postdoctoral Fellowship Program. H.D. and M.Z. were supported by the Sino-Norway Collaborative STRESSOR Project (NSFC, Grant No. 41861134040) and also acknowledge support from the Shanghai Frontiers Science Center of Polar Science (SCOPS). H.D., M.Z., and R.R. were all supported by the Dragon 6 Program (Grant No. 95451). R.R. was additionally supported by the PRODEX Program of the European Space Agency (S23DEddy, Grant No. 4000135226). F.O. acknowledges support from CNES (project BIOSWOT-ADAC). I.F. acknowledges support from the Research Council of Norway through the PROVOLO Project (Grant No. 250784) and thanks Dr. Anthony Bosse for his contribution to glider data processing. L.S. was supported by NASA (Grant No. 80NSSC24K1653). All authors acknowledge the Norwegian facility for Ocean Gliders (NorGliders) at the University of Bergen for providing access to the Seaglider dataset. We are also grateful to Drs. Xiaodong Wu, Rufu Qin, Yong Zhuang and Shengyang Fang for valuable discussions, and to the four anonymous reviewers and the editor for their constructive comments.

## Author contributions

H.D. and M.Z. conceived and design this study. H.D. performed the analyses, visualization, and drafted the manuscript. J.M. and F.O. contributed to the analyses and interpretation. R.R. conducted the mesoscale eddy detections, and I.F. led the ocean glider program and contributed to the glider data processing. L.Q., B.Q., L.S., Z.Z., and W.S. reviewed and improved the manuscript. A.K.S. executed the high-resolution simulation. All authors discussed the results and contributed to the final manuscript.

## Competing interests

The authors declare no competing interests.
