## [Transparent Peer Review file · Nature Communications]

Warm Rings in Mesoscale Eddies in a Cold Straining Ocean

Corresponding Author: Dr Huizi Dong

Version 0:

Reviewer comments:

Reviewer #1

(Remarks to the Author)

The manuscript describes the vertical heat transport at the skirts of mesoscale cyclones and anticyclones, suggesting a unified mechanism of upwelling-downwelling system. The mechanism suggested consists in an ageostrophic cross-front motion of fluid parcels of different direction at the upper and lower levels, which induces upwards/downwards vertical motions along the boundaries of the mesoscale eddies. These motions were suggested to be formed (or intensified?) episodically during the periods of an increased strain at the eddy boundaries, which, in turn, were a result of mesoscale eddy interactions with submesoscale structures. The resulting heat flux is speculated to be responsible for formation of the observed warm-core rings around both cyclones and anticyclones, as well as for enhancing the resulting heat flux from the ocean to the atmosphere.

The study is based on high-resolution Seaglider data in the Lofoten Basin, combined with the satellite altimetry data and high-resolution SST data. Eddies were derived by a threshold method using the value of Okubo-Weiss parameter in the closed SLA. Although this method is not optimal, it allows detecting strong mesoscale eddies at the sea-surface. The authors present a novel concept, which is supported with observations. The manuscript represents an interesting study and is well written. I recommend publication after minor modifications.

The explanation of the mechanism is vague. More details on how the sharpening of the front result in ageostrophic vertical motions would be beneficial. Are there any theoretical estimates on the intensity of such process? These can be compared with the derived vertical velocities.

There are also two additional points, which might be noted in the manuscript.

The authors should also mention, that the suggested mechanism is not the only one which can form such warm-ring structures. Another mechanism, acting in particular for undersurface anticyclones is trapping of a warmer water and wrapping it around the eddy center. In particular, this was observed for surface manifestation of meddies and the Lofoten Vortex – see Bashmachnikov et al., 2013. Manifestation of two meddies in altimetry and sea-surface temperature; Bashmachnikov et al., 2018. Pattern of vertical velocity in the Lofoten vortex (the Norwegian Sea. This forms far larger SST anomalies around eddies compared to 0.4°C (only slightly above on the limit of accuracy of the IK SST sensors) of the mechanism suggested by the authors.

The vertical velocities at the eddy boundaries, suggested by the authors (Figure 6b), do not fully merge with the vertical velocity structure in the Lofoten Vortex, derived using MIT model in the paper cited above. In the Lofoten Vortex the upwelling was dominating all the central part of the vortex and downwelling was observed at its outer skirt, but these motions were characteristics only at the level of the well mixed eddy core, intensifying towards the sea-surface. The driving mechanism was suggested to be also linked to a high strain at the eddy boundary, which increases towards the sea-surface. The resulting decay of kinetic energy form in the upper ocean an inverse secondary circulation, similar by nature to the one formed by the bottom Ekman boundary layer (see also Bosse et al., (2019) Dynamical controls on the longevity of a non-linear vortex: The case of the Lofoten Basin Eddy,). This is another effect the strain, and the resulting vertical pattern should merge with the one suggested in the manuscript. Both mechanisms should form increase their intensity towards the sea-surface, where the largest energy (straining) of submesoscale is expected. This can be discussed and noted at the sketch in Figure 6b.

Minor comments:

49-50 “The North Atlantic Current is an important part of the oceanic conveyor belt that transports heat from the equator to the poles.” The NAC, as well as the Global Conveyor, does not reach the poles – please rephrase.

51-53 "the Atlantic Water experiences substantial heat loss to the atmosphere, which is thought to be the main driver of the transformation of Atlantic Water" - one may also see Dugstad et al (2018) "Lateral Heat Transport in the Lofoten Basin: Near-Surface Pathways and Subsurface Exchange" and Bashmachnikov et al., 2023 "Heat transport by mesoscale eddies in the Norwegian and Greenland seas" on the role of mesoscale eddies in regulating the AW transport across the Lofoten Basin. 53-55 "This process significantly impacts the hydrographic environment, sea ice cover and ecosystems in the region and ultimately the Arctic Ocean" – What process from the mentioned above? The impact of AMOC on SST in the Nordic Seas is nonlinear and involves the atmospheric response to the ocean heating (Bengtson et al., 2004 "The Early Twentieth-Century Warming in the Arctic—A Possible Mechanism", Shaffrey and Sutton, 2006 "Bjerknes Compensation and the Decadal Variability of the Energy Transports in a Coupled Climate Model")

60 Eddy (LBE), which subsequently is stabilized by bottom topography – the more appropriate citations are Kohl 2007 "Generation and stability of a quasi-permanent vortex in the Lofoten Basin", Bosse et al., (2019) Dynamical controls on the longevity of a non-linear vortex: The case of the Lofoten Basin Eddy, Santieva et al., 2021 "On the stability of the Lofoten Vortex in the Norwegian Sea"

157 How this critical point is defined?

165-166 "further indicating the presence of a cross-frontal secondary circulation" - Small Ri does not necessarily mean enhanced vertical velocities. The facilitated Kelvin-Helmholtz instability does not require the cross-frontal exchange.

168 What type of ageostrophic disturbances is mentioned here? Linked to an enhanced vertical shear?

209 LCS – abbreviation is not presented

212-216 "the scale across the front becomes small enough and the geostrophic balance can no longer be maintained" What do you mean by "small enough"? Less than Rossby radius of deformation? How this decreasing the cross-front scale affects the geostrophic balance?

545-547 Ri characterizes stability of the vertically sheared flow. $Ri \gg 1$ means that Kelvin-Helmholtz instability cannot be generated. I do not see the direct relation between Ri and geostrophic-ageostrophic separation, which is basically linked to the Rossby number.

Reviewer #2

(Remarks to the Author)

As requested by the editor of Nature Communications, I reviewed the manuscript by Dong et al., titled "Warm rings in mesoscale eddies in a cold straining ocean". This study provides a thorough analysis of the vertical heat transport associated with submesoscale filaments around mesoscale eddies in the Lofoten Basin of the North Atlantic, and the resulting formation of surface warm ring structures, based on long-term SeaGlider observation data. I believe it is rare to find such a beautiful visualization of heat advection by submesoscale structures from actual ocean observations. Therefore, I support its publication in Nature Communications. However, there are several critical concerns described below. Please clarify these points.

Major concerns:

1. TS Distribution & Salinity Contribution:

- Request to see the TS distribution that determines the density of seawater.
- Accurate quantification of salinity's contribution to density and geostrophic flow is needed.
- Figure 6b implies that temperature is dominant, but in high-latitude regions (70-72°N), salinity typically plays a more significant role in density-driven flows (e.g., Rudels et al. 2013, doi:10.1016/j.pocan.2013.11.006).

2. Definition of "Edge" (L92, L98):

- The definition of "edge" is unclear throughout the paper.
- The use of " $r = R$ " to describe the "eddy edge" is questionable. A clear definition of R is needed, possibly the location of maximum azimuthal velocity.

3. Relative Vorticity & Rankine Vortex (L463):

- Request to see the spatial distribution of relative vorticity (ζ) in eddy-centric coordinates.
- Suggest assuming a Rankine vortex profile ($\zeta = \text{constant}$ for inner regions; zero for outer regions) to clarify the positioning of R .

4. Geostrophic Velocity Resolution:

- Questioning whether the geostrophic velocity (u_g, v_g) can be truly resolved from sea surface height (SLA).
- Concern about whether the satellite data resolution (0.25 x 0.25 degrees, ~27 km latitudinal resolution) is sufficient to resolve the eddy's horizontal structure, given the deformation radius (~10 km).

Specific comments:

L31-32: having "poleward" appear twice in the same sentence feels repetitive. You should rephrase the sentence to avoid the redundancy. Like "heat loss during its transit towards the polar regions."

L60: Clarify if "stabilized" means the eddy disappears. Consider using "weakened" or "dissipated."

L67: Note that heat diffusion around polar eddies also involves mechanisms like double diffusion or mixing by internal waves (e.g., Kawaguchi et al. 2012, doi:10.1016/j.dsr.2012.04.006; Kawaguchi et al. 2016, doi:10.1175/JPO-D-15-0150.1).

L71: The origin of the 0.1-10 km range is unclear. The upper bound should relate to the deformation radius.

L78: "frontogenesis" should not be capitalized.

L93: A description of the TS structure of cyclones and anticyclones below the mixed layer is necessary. Mention that salinity contributes minimally to density-driven GQ flow.

L115: The term FSILE should be defined upon its first mention.

L123: Provide a simplified definition of Ri or reference the relevant Methods section.

L176: What is the relative contribution of temperature and salinity to restratification?

L201: Clarify the "hyperbolic region along the edges." Is it inside or outside?

L209: Define "LCS."

L213: The statement "frontogenesis develops by releasing the available potential energy of the front" might be inaccurate. If internal density surfaces stand vertically, wouldn't APE increase?

L216: The "warm side to cold side" assumption only applies if the density profile is temperature-driven. In polar oceans, this could be reversed. Start by explaining the TS structure inside and outside the eddy.

L220: Similar clarification is needed: "Warm fluids are located within the anticyclonic eddy, whereas cold fluids are ..."

L474: The dimensions of " δ_i " and " δ_f " (set at 0.02° and 0.4°) are unclear. Are these distances and assumed as constants? Also please clarify the independent variable in equation (1).

L529: Clarify whether "s is the along-track direction" means distance instead of "direction."

L542: The use of "N" for buoyancy frequency and "N" for sampling number (Fig. 3) overlaps. Change one of the notations.

L545: Ri's formula should be presented separately and referenced earlier in the text.

L577: "anomalies" should be clarified— anomalies relative to what?

L579: Use consistent notation for $|b_x|$ —is it b_x or B_x ?

L587: Provide the definition of R_x .

L590: Clarify "position relative to the eddy" as being either inside or outside relative to the eddy's core.

Figures and captions:

Figure 1 (L288, L291, L295):

- Blue dots on the blue background are hard to see; change the color of the background or the dots.
- Provide full forms of SST, CHL, and FSLE for clarity.
- Orange dots on the red background are difficult to see.
- In panel c, showing a vertical section of temperature and salinity from the gliders (even in the supplementary material) would be helpful.

Figure 2 (L306):

- The notation for "Bx" seems inconsistent. Check whether "Bx," "B_x," "b_x," and "bx" are mixed up.

Figure 3 (L321):

- The panel order in the caption is inconsistent. Either follow alphabetical order or change the panel layout for clarity.
- Consider separating the left and right groups of panels with space and make the variables (Strain, Ri, SSTa) more prominent on the right.
- The definition of "R" as the normalized radius of the eddy is unclear. Is R the point where azimuth velocity becomes maximal?
- The notation for "N" and "N*" in panels a and b should be changed to avoid confusion with buoyancy frequency.

Reviewer #3

(Remarks to the Author)

The study uses glider observations near eddies to investigate vertical heat transport (VHT) driven by submesoscale processes down to 400 meters, with claims that this deep-reaching VHT warms the upper ocean and contributes to warm eddy rings. While the topic is promising, the manuscript currently suffers from significant issues in methodology, theoretical foundation, and dynamical interpretation. The authors seem to have limited experience with glider observations and an incomplete understanding of geophysical fluid dynamics, especially concerning high-frequency, small-scale ocean processes. Given these gaps, I believe that substantial portions of the study should be restructured, potentially with greater reliance on model results for a more reliable analysis.

In its first part, the study largely repeats findings from Siegelman et al. (2020), employing similar data but in a different region, with fewer mechanisms discussed. Scientific innovation diminishes when a concept is reiterated without substantial new insights; hence, this section lacks impact. The second part includes a composite analysis describing “warm rings” within eddies, a feature well-documented in previous literature. Although the authors attribute these rings to submesoscale VHT, there is no direct evidence linking the rings specifically to submesoscale rather than mesoscale dynamics, making the analysis questionable.

A critical flaw of the method is the observing platforms. In the past two decades, observations using gliders do improve the observation skills especially about meso- and smaller-scale physics. Gliders may observe mesoscale features, but gliders, with a slow movement of approximately 10 km/day, cannot adequately resolve lateral submesoscale features. As established by Rudnick and Cole (2011, doi:10.1029/2010JC006849), the Doppler smearing and aliasing effects in glider data lead to significant errors in the submesoscale range, particularly in stratified layers. I think the author need to carefully go through this paper before working on this study. In fact, Studies typically use gliders for vertical profiles or along-isopycnal analyses, or in networked arrays to address resolution issues (e.g., Rudnick et al. 2022, doi:10.1175/JPO-D-21-0181.1). Consequently, I find the buoyancy gradients presented in Fig. 1e unreliable, which casts doubt on the Richardson number (Ri) and vertical velocity (w) estimates. The results of VHT are quite questionable unless the authors can demonstrate that the smearing and aliasing effects have limited impacts. As suggested in the recent several field campaigns these years, observations using onboard towed bodies (with ship speed >6 knots) are preferable to minimize such errors, although this is challenging.

What do you mean by 3D view of vertical velocity (Line 161 and Fig. 4). The results are just presented without any clear interpretation. The results lack clear interpretation, and it's unclear if they include submesoscale effects. If they do, this would be unexpected. If derived from composite fields, these vertical velocity estimates are likely inaccurate. Using composite eddy fields to infer ageostrophic processes is misleading, as this approach fails to capture true dynamics. It's important to note that the composite field approach is essentially a method to illustrate the general features of the eddy and should not be used to quantify submesoscale characteristics. Composite fields average over time and space, which tends to smooth out smaller-scale variations and can misrepresent the true intensity and structure of submesoscale processes. Therefore, using composite fields to infer submesoscale dynamics is not methodologically sound and likely introduces significant inaccuracies.

Additionally, the role of mesoscale contributions to VHT is underrepresented. Mesoscale processes likely contribute substantially to VHT, especially in an averaged context, and should be acknowledged.

More calibration details on the glider data are needed, especially given the extended observation period.

The sentence on Line 157, “When the scale across the front is compressed to a critical point, geostrophic balance is no longer maintained. By breaking the geostrophic balance that suppresses vertical motions, the fronts support vertical velocities of up to 60 m day⁻¹ below the surface mixed layer,” is confusing. The breakdown of geostrophic balance typically increases vertical motions via non-geostrophic terms, rather than suppressing them.

In summary, the findings presented here are largely redundant with existing literature or are based on analysis and interpretation that lack sufficient rigor. The data-supported results largely replicate previously reported findings, while the remaining conclusions appear to stem from a lack of careful interpretation and logical coherence. This leaves an impression of incomplete novelty and unsound analytical approaches within the study.

Version 1:

Reviewer comments:

Reviewer #1

(Remarks to the Author)

I think the authors has done a perfect work in revising the manuscript and it is now ready for publication.

Reviewer #2

(Remarks to the Author)

Please refer to the separately uploaded letter for the comments.

[Editorial Note: This letter has been appended to the end of this file]

Reviewer #3

(Remarks to the Author)

I appreciate the substantial revisions made in this version of the manuscript. The authors have addressed some of my previous concerns; however, several major issues remain unresolved. These concerns are critical and may introduce significant uncertainties in the results presented.

1. On the novelty of the findings. While the authors continue to emphasize a connection between warm rings and mesoscale eddies, the core of the study essentially reflects enhanced vertical heat transport at oceanic fronts, as comprehensively addressed in Siegelman et al. (2020) and Cao et al. (2024) where similar dynamics and methodologies are presented. By the way, I think wrong citation for Cao et al. (2024) is listed in the reference. Upon revisiting these two references, I find that both the underlying dynamics and diagnostic methods have already been thoroughly explained in the existing literature. The current manuscript does not appear to introduce substantial new insights beyond these established findings. Although the authors attempt to highlight the elevated values of vertical heat transport (VHT), this alone does not convince me as a significant novelty, especially considering that these values may be overestimated, as discussed below.

2. Concerns regarding data contamination. The issue of potential data contamination remains insufficiently addressed. While the authors have now described stricter data quality control procedures and implemented a 10-km smoothing filter to mitigate aliasing and smearing effects, there is no clear evidence demonstrating the effectiveness of this approach. A straightforward way to validate the impact of filtering may be to compare the wavenumber spectra before and after applying the filter. Based on my own experience with glider data, this method is unlikely to fully resolve contamination at submesoscales. It appears the authors may have misunderstood the findings of Rudnick et al. (2011): the primary limitation arises from the slow pace of gliders, not the quality of the sensors. Temporal and spatial glider motion can cause substantial aliasing from internal waves, limiting the reliable resolution of submesoscale variability. While gliders are valuable tools for oceanographic observations, their use in submesoscale studies carries inherent challenges that must be more carefully considered here.

3. Concerns about composite analysis methodology. I continue to believe that the composite analysis employed is misleading. The mesoscale eddy structures included are highly variable, and their differences cannot be adequately captured by composite means. Critically, the strength and structure of fronts appear to be the primary drivers of VHT, as shown in Siegelman et al. (2020). These frontal characteristics are in turn influenced by external forcing, background flow, and possibly eddy–eddy interactions. The manuscript does not sufficiently define or justify how azimuthal variability is treated or interpreted across the ensemble of eddy cases. Given this heterogeneity, I strongly recommend that the authors instead present representative vertical sections from selected cases, which would offer a more physically meaningful and interpretable analysis.

Version 2:

Reviewer comments:

Reviewer #3

(Remarks to the Author)

Many of previous have examined vertical heat transport associated with submesoscale processes near the fronts using both models and observations. The topic is not new. As highlighted by the authors, the main novelty of this manuscript lies in the duration of the dataset and the quantification of VHT. However, the glider observations employed here are inherently limited in their ability to accurately resolve submesoscale dynamics, due to their sampling technique. This limitation introduces substantial uncertainties in studying submesoscales, which are not adequately addressed in the current study. In addition, Many of the assumptions used in this study lack justification, significantly undermining the reliability of the conclusions.

In this revision, the authors have presented some additional analyses in response to my earlier concerns. However, these additions do not fully resolve the issues I raised. It is important for the authors to recognize that glider data have been available for over two decades, and yet they are seldom used to study submesoscale dynamics. Despite the apparent presence of submesoscale features, the slow sampling rate of gliders introduces significant contamination, particularly at scales below 30 km, often resulting in large errors, typically overestimations.

I carefully reviewed the method proposed by the authors to mitigate this contamination, which involves smoothing the isopycnal depths. This approach is interesting and does reduce spectral variance substantially almost at all scales and especially at smaller scales. However, assuming the method is effective, the manuscript does not show any significant change in results. In this point, I seriously question the authors' academic rigor. The response appears more focused on persuading the reviewer than on demonstrating robust, justified scientific results.

Moreover, the application of a 10-km low-pass filter inevitably removes a substantial portion of the submesoscale signal. why a 10-km threshold is used? Reconstructing T and S fields based on smoothed isopycnal surfaces does not guarantee the removal of internal wave signals. The approach seems functionally similar to applying a 25-hour temporal smoothing, which also reduces internal wave influence but does not eliminate contamination from aliasing. This would also remove high-frequency submesoscale motions. In this way, the primary effect appears to alter the buoyancy gradient and T' , both of which are critical to the estimation of VHT. If the authors are confident in the validity of their approach, I recommend they simulate glider sampling using model output to test its reliability. From experience, such simulations often yield results that diverge considerably from expectations.

Once again, I insist that the composite-based quantification of submesoscale-driven VHT presented here is definitely misleading when based on glider data. In reality, gliders may move at speeds comparable to mesoscale eddies, capturing only a limited and potentially unrepresentative portion of the eddy structure, let alone the submesoscale variability. The

analysis relies on the assumption that mesoscale eddies are stationary and steady, which is not entirely accurate. While such an assumption may be acceptable for identifying general eddy structures, it is not appropriate for quantifying specific variables like submesoscale-driven VHT. With the advent of new satellite missions like SWOT, we are beginning to observe significant asymmetries in mesoscale eddies, suggesting that many conventional assumptions may be ill-used and could lead to large errors in quantification. It is not a good time to go back.

Reviewer #4

(Remarks to the Author)

The manuscript titled “Warm Rings in Mesoscale Eddies in a Cold Straining Ocean” provides a well written report on submesoscale heat transport in the Lofoten Basin and reveals novel insights into the dynamical regimes present. The results have been presented in a clear and concise manner, and the analysis is of a sound standard. The primary purpose of my review is to provide additional guidance on the whether concerns raised by Reviewer 3 have been sufficiently addressed. I believe these points have been largely resolved, but minor deficiencies remain in the discussion of smearing and aliasing effects. My review reflects on the response to each of Reviewer 3’s points in turn, followed by two additional (minor) suggestions.

Point 1 – The authors raise valuable points in response to the assertion that the manuscript lacks novelty and I believe the paper is of significance to the field. However, the narrative in some places leads the reader to conclude that the present study is the first of its kind, when the headline result is building on that of Siegelman et al. (2020). Although not critical for publication, some minor changes to the text may help differentiate the manuscript’s novel findings from that of existing literature. My main recommendations are that (1) the study’s isolation of the cyclonic and anticyclonic contributions to submesoscale vertical heat transport is made clearer on lines 106-108 and (2) the framing of the text between lines 284-290 is refined to acknowledge that submesoscale heat fluxes in regions of strain has been identified in previous studies.

Point 2 – The authors have made considerable efforts to respond to the Reviewer 3’s concern on data contamination. In my view, their expansion of the analysis to compare spectral decompositions and analysis of the impact of isopycnal filtering sufficiently addresses the reviewers concern around the lack of transparency in qualifying the use of glider observations for the manuscript’s purposes. Results showing that fine-scale features remain in the buoyancy gradient and vertical velocity fields upon isopycnal filtering is sufficient indication that these are associated with submesoscale dynamics. The authors have taken on board the reviewer’s suggestions, but in the process have omitted a key distinction between smearing and aliasing that would aid their case. For example, line 305 quotes “Doppler aliasing” when this should be “Doppler smearing”. As outlined by Rudnick et al. (2011), aliasing is a function of sampling rate and Doppler smearing is associated with the speed that the observing platform travels through the water. As a minimum, I would recommend that a distinction is made in the text between the source of aliasing and smearing, and that the authors are careful not to use the terms interchangeably. Furthermore, though not necessary for publication, it could benefit the manuscript to cite the relevant length and spatial scales associated with these effects (e.g., Rudnick et al., 2011). It is possible to estimate the spatial- and temporal-scales upon which smearing and aliasing act, and the sampling intervals cited in the supplementary material indicate there may be limited aliasing at the submesoscale in the observations in question.

Point 3 – The authors have performed the additional analysis requested by Reviewer 3 regarding the composite analysis and their concerns have been addressed to a satisfactory level. Despite significant non-uniformity in eddy structure, the composite analysis appears to provide a robust measure of the alignment between vertical heat transport and strain.

Additional Suggestions

The sentence starting “The rotational flow...” on lines 279-282 is difficult to interpret. As far as I can tell, the intention is to convey that eddies can advect water masses. Since this is already stated in the previous sentence, I would suggest this sentence is removed. In addition, Reference 35 does not appear to have relevance to the claims made and, should the sentence remain, I would suggest reviewing whether this citation is appropriate.

Line 287 currently reads “eddy eddies”. Should this be “eddy edges”?

I hope this feedback is received as constructive and I look forward to seeing the work published, should that be the final decision.

Kind Regards,
Dr. R. D. Patmore (NOC, Liverpool, UK)

Version 3:

Reviewer comments:

Reviewer #4

(Remarks to the Author)

My previous concerns have been addressed to a satisfactory level and I have no further feedback for the authors. I

appreciate their efforts to improve the manuscript.

Response to Review Comments for

“Warm Rings in Mesoscale Eddies in a Cold Straining Ocean”

Huizi Dong, Meng Zhou, James C. McWilliams, Roshin P. Raj, Francesco d’Ovidio, Ilker Fer, Lixin Qu, Bo Qiu, Lia Siegelman, Zhengguang Zhang, Walker O. Smith, Jr., Ann Kristin Sperrevis

Note: Reviewer’ comments are in italic font; authors’ response comments are in normal font. Revisions in the revised manuscript are highlighted. Figs. 1–6 are presented in the revised main manuscript; Figs. S1–S9 are included in the Supplementary Information; and Figs. R1–R2 are provided in this response document.

REVIEWER COMMENTS

Reviewer #1 (Remarks to the Author):

The manuscript describes the vertical heat transport at the skirts of mesoscale cyclones and anticyclones, suggesting a unified mechanism of upwelling-downwelling system. The mechanism suggested consists in an ageostrophic cross-front motion of fluid parcels of different direction at the upper and lower levels, which induces upwards/downwards vertical motions along the boundaries of the mesoscale eddies. These motions were suggested to be formed (or intensified?) episodically during the periods of an increased strain at the eddy boundaries, which, in turn, were a result of mesoscale eddy interactions with submesoscale structures. The resulting heat flux is speculated to be responsible for formation of the observed warm-core rings around both cyclones and anticyclones, as well as for enhancing the resulting heat flux from the ocean to the atmosphere. The study is based on high-resolution Seaglider data in the Lofoten Basin, combined with the satellite altimetry data and high-resolution SST data. Eddies were derived by a threshold method using the value of Okubo-Weiss parameter in the closed SLA. Although this method is not optimal, it allows detecting strong mesoscale eddies at the sea-surface.

The authors present a novel concept, which is supported with observations. The manuscript represents an interesting study and is well written. I recommend publication after minor modifications.

1) The explanation of the mechanism is vague. More details on how the sharpening of the front result in ageostrophic vertical motions would be beneficial. Are there any theoretical estimates on the intensity of such process? These can be compares with the derived vertical velocities.

Response: We appreciate the comments and suggestions made by the reviewer. To address the concern regarding the previously vague explanation of how frontal sharpening leads to ageostrophic vertical motions, we have made the following improvements:

- The original unclear statements have been modified to “As the cross-front scale is compressed by the strain field, frontogenesis intensifies, increasing the buoyancy gradient and ultimately disrupting the thermal wind balance. To restore this balance, an ageostrophic secondary circulation develops, characterized by ...” in Lines 171-176 of the revised manuscript.
- We have been clarified the explanation “This occurs because, irrespective of being cyclonic or anticyclonic, the submesoscale secondary circulations act to flatten sloping isopycnals at the fronts, thus restoring stratification...” in Lines 207-210 of the revised manuscript.
- The original sentences have been modified to “This is attributed to frontogenesis, whereby strong geostrophic strain intensifies lateral buoyancy gradients along the eddy edges...” in Lines 231-233 of the revised manuscript.

In response to the reviewer’s suggestion regarding theoretical estimates of this process and their comparison with observed vertical velocities:

- We have added a section "Quasi-geostrophic frontogenesis theory and vertical velocity" to the Supplementary Information (Lines 33–69).

There are also two additional points, which might be noted in the manuscript.

2) The authors should also mention, that the suggested mechanism is not the only one which can form such warm-ring structures. Another mechanism, acting in particular for undersurface anticyclones is trapping of a warmer water and wrapping it around the eddy center. In particular, this was observed for surface manifestation of meddies and the Lofoten Vortex – see Bashmachnikov et al., 2013. Manifestation of two meddies in altimetry and sea-surface temperature; Bashmachnikov et al., 2018. Pattern of vertical velocity in the Lofoten vortex (the Norwegian Sea). This forms far larger SST anomalies around eddies compared to 0.4oC (only slightly above on the limit of accuracy of the IK SST sensors) of the mechanism suggested by the authors.

Response: We agree with the reviewer that submesoscale motions are not the only mechanism for the formation of warm ring structures. We have incorporated the suggested essential contents in Lines 275-285 and Lines 292-300 of the revised version. Additional mechanisms related to warm rings and mesoscale vertical motions have also been supplemented in the main text and Supplementary Information.

To further clarify for the reviewer the critical role of submesoscale vertical motions at eddy edges in the northern Norwegian Sea, which is our area of interest, we would like to provide a supplementary case analysis from an alternative perspective. Fig. R1 presents an additional case study based on Seaglider data from May 2017. This section was specifically chosen because the Seaglider crossed the edge of the same anticyclonic eddy three times during the one-month period (Fig. R1b). Corresponding nutrient and chlorophyll profile observations (Fig. R1a,b), obtained from the NMDC database¹ (<https://doi.org/10.21335/NMDC-1271328906>) during the same period, allowed us to analyze vertical transport patterns at both eddy-edge and non-edge regions.

The observations revealed pronounced lateral buoyancy gradients and strong vertical velocities at the eddy edges, which facilitated enhanced upward transport of nutrients (e.g., nitrate, silicate) into

the surface layer. The increased surface nutrient concentrations subsequently led to elevated chlorophyll-a concentrations, providing indirect but compelling evidence of intensified submesoscale-driven vertical exchange at the eddy edges (Figs. R1g–i).

Similar results from additional cases involving both anticyclonic and cyclonic eddies (Fig. R2b–c) consistently show that submesoscale vertical circulation at eddy edges significantly enhances nutrient availability and productivity relative to eddy interiors and external waters. This body of evidence collectively demonstrates that submesoscale-driven vertical transport mechanisms play a dominant role in the formation of warm-ring structures at eddy edges.

Fig. R1 | Characteristics of submesoscale motions of anticyclonic eddy edges and their impacts on vertical nutrient distribution. **a** SLA and **b** FSLE fields overlaid with the Seaglider track from 7 May 2017 to 2 June 2017. The Seaglider traveled approximately 450 km from inside the anticyclonic eddy

(point S1 in panel **a**) to outside the eddy (point S2), with measurements shown in panels **c-g**. The glider's vertical sections show **c** buoyancy, **d** lateral buoyancy gradient (Bx), **e** vertical velocity (w) and **f** Richardson number (Ri). The mixed layer depth (MLD) is shown by a thick black line in **c-f**. Yellow and green stars in panels **a** and **b** represent BGC stations at the eddy edge and non-edge locations, respectively, on 22 May 2017. **g-i** Vertical profiles of nitrate, silicate and chlorophyll-a (CHL-a) concentrations, where red lines represent profiles at the eddy edge station (yellow star in panel **a**) and black lines show profiles at the non-edge station (green star).

Fig. R2 | Vertical distribution of nutrients and chlorophyll in edge and non-edge regions of two anticyclonic eddies and one cyclonic eddy. a-c FSLE and SLA fields for three mesoscale eddy cases on a 22-May-2017, **b** 13-June-2022, and **c** 23-July-1995, overlaid with BGC stations at the eddy edge (yellow stars) and non-edge locations (green stars). **d-f** Vertical profiles of nitrate, silicate and chlorophyll-a (CHL-a) concentrations, where red and blue lines represent profiles at the eddy edge station (yellow stars in panels **a-c**) and black lines show profiles at the non-edge station (green stars in panels **a-c**).

3) The vertical velocities at the eddy boundaries, suggested by the authors (Figure 6b), do not fully merge with the vertical velocity structure in the Lofoten Vortex, derived using MIT model in the paper cited above. In the Lofoten Vortex the upwelling was dominating all the central part of the vortex and downwelling was observed at its outer skirt, but these motions were characteristics only at the level of the well mixed eddy core, intensifying towards the sea-surface. The driving mechanism was suggested to be also linked to a high strain at the eddy boundary, which increases towards the

sea-surface. The resulting decay of kinetic energy form in the upper ocean an inverse secondary circulation, similar by nature to the one formed by the bottom Ekman boundary layer (see also Bosse et al., (2019) Dynamical controls on the longevity of a non-linear vortex: The case of the Lofoten Basin Eddy). This is another effect the strain, and the resulting vertical pattern should merge with the one suggested in the manuscript. Both mechanisms should form increase their intensity towards the sea-surface, where the largest energy (straining) of submesoscale is expected. This can be discussed and noted at the sketch in Figure 6b.

Response: We agree with the reviewer that the circulation pattern of the LBE is indeed driven by multiple mechanisms. As the reviewer noted, due to differences in model resolution, the vertical velocity structure at the eddy edges in our study differs from that derived from the 4 km-resolution MITgcm model used by Bashmachnikov et al. (2018). Bashmachnikov et al. successfully reproduced the radius and depth of the LBE and described vertical velocity patterns both within and around the LBE. Our higher-resolution Seaglider observations (horizontal resolution of approximately 1–4 km, averaging ~2 km at eddy edges) build upon this work by providing a more detailed view of finer-scale vertical dynamics, especially the strong lateral buoyancy gradients and intensified strain at the eddy edges. Specifically, the vertical velocities we observed within the eddy interiors were consistent with those reported by Bashmachnikov et al., while the velocities at the eddy edges were one to two orders of magnitude higher. Thus, while multiple mechanisms contribute to the LBE circulation, our results suggest that submesoscale processes play a particularly prominent role in vertical transport.

To further clarify the relative contributions of submesoscale and mesoscale processes, we have added a section titled “Comparison between mesoscale and submesoscale VHT” in the Supplementary Information (Lines 148–177 of the revised manuscript). Bosse et al. (2019) similarly highlighted that the 4-km-resolution numerical model underestimated the maximum azimuthal velocity and core vorticity, suggesting that submesoscale processes could be significant at eddy edges, thus requiring higher-resolution simulations. To address this, we have strengthened our analysis using the high-resolution Norkyst-DA model (2.4 km horizontal resolution) in the revised version. Two case studies from this model demonstrate similarly strong vertical velocities at eddy edges (Lines 152-158). Furthermore, composite analysis of model outputs (2017-2019) reveals that intense submesoscale VHT predominantly occurs at eddy edges, while mesoscale VHT, though more widely distributed, is approximately an order of magnitude weaker. Even without accounting for net heat fluxes, mesoscale VHT magnitude represents only about 1/6 to 1/5 of the submesoscale contribution (see "Numerical simulation verification" section in the Supplementary Information).

Regarding the reviewer's suggestion to add annotations to Fig. 6, considering that our study focuses not only on the LBE but also on eddies across the eastern basin and continental slope regions (Fig. 1a), we chose to include the suggested clarifications in the main text (Lines 277-286 and 292-300)

rather than modifying Fig. 6. This approach preserves the schematic's broad applicability regarding submesoscale secondary circulations at eddy edges as presented in Fig. 6.

Minor comments:

49-50 *“The North Atlantic Current is an important part of the oceanic conveyor belt that transports heat from the equator to the poles.” The NAC, as well as the Global Conveyor, does not reach the poles – please rephrase.*

Response: Following the reviewer’s suggestion, we have revised the text to ‘The North Atlantic Current is an important part of the oceanic conveyor belt that transports heat from the equator to the high latitudes’ in Lines 51-52 of the revised manuscript.

51-53 *“the Atlantic Water experiences substantial heat loss to the atmosphere, which is thought to be the main driver of the transformation of Atlantic Water” - one may also see Dugstad et al (218) “Lateral Heat Transport in the Lofoten Basin: Near-Surface Pathways and Subsurface Exchange” and Bashmachnikov et al., 2023 “Heat transport by mesoscale eddies in the Norwegian and Greenland seas” on the role of mesoscale eddies in regulating the AW transport across the Lofoten Basin.*

Response: We have revised the text to ‘the Atlantic Water undergoes transformation driven by both substantial heat loss to the atmosphere^{4,5} and lateral heat transport through mesoscale eddies^{6,7}’ in Lines 54-55 of the revised version. The suggested references have been added accordingly in the revised version.

53-55 *“This process significantly impacts the hydrographic environment, sea ice cover and ecosystems in the region and ultimately the Arctic Ocean” – What process from the mentioned above? The impact of AMOC on SST in the Nordic Seas is nonlinear and involves the atmospheric response to the ocean heating (Bengtson et al., 2004 “The Early Twentieth-Century Warming in the Arctic— A Possible Mechanism”, Shaffrey and Sutton, 2006 “Bjerknes Compensation and the Decadal Variability of the Energy Transports in a Coupled Climate Model”)*

Response: We have revised the text to ‘This transformation of Atlantic Water significantly impacts the hydrographic environment, sea ice cover, and ecosystems in the region and ultimately the Arctic Ocean’ in Lines 55-57 of the revised version.

60 *Eddy (LBE), which subsequently is stabilized by bottom topography – the more appropriate citations are Kohl 2007 “Generation and stability of a quasi-permanent vortex in the Lofoten Basin”, Bosse et al., (2019) Dynamical controls on the longevity of a non-linear vortex: The case*

of the Lofoten Basin Eddy, Santieva et al., 2021 “On the stability of the Lofoten Vortex in the Norwegian Sea”

Response: We have updated Line 64 with the more appropriate citations (Kohl et al., 2007; Bosse et al., 2019; Santieva et al., 2021) in the revised version.

157 How this critical point is defined?

Response: We agree with the reviewer that the term ‘critical point’ was ambiguous. We have expanded the paragraph as follows: “As strain fields compress the front, the cross-frontal buoyancy gradient intensifies and the cross-frontal scale decreases, driving frontogenesis and increasing the available potential energy at the front. When the local Rossby number ($R_o = \zeta/f$) approaches $O(1)$, thermal wind balance becomes disrupted, triggering ageostrophic secondary circulations that act to restore this balance. These circulations generate upwelling on the warm (light) side and downwelling on the cold (heavy) side of the front.” in Lines 250-256 of the revised manuscript.

165-166 “further indicating the presence of a cross-frontal secondary circulation” - Small Ri does not necessarily mean enhanced vertical velocities. The facilitated Kelvin-Helmholtz instability does not require the cross-frontal exchange.

Response: We appreciate the reviewer's comments, which highlight an important clarification. Here, we calculated the geostrophic Richardson number ($Ri = f^2 N^2 / b_x^2$), which characterizes the stability of the thermal wind balance under intensified buoyancy gradients (frontogenesis). A low geostrophic Ri value indicates that the thermal wind balance is disrupted by strong lateral buoyancy gradients, thereby directly facilitating the development of cross-frontal ageostrophic secondary circulations. We have clarified this in the manuscript (Lines 197-201) and included the formula when Ri is first mentioned (Line 133) in the revised manuscript.

168 What type of ageostrophic disturbances is mentioned here? Linked to an enhanced vertical shear?

Response: The sentence has been revised in Lines 198-201 to read “These results are consistent with the expectation that geostrophic strain shrinks the frontal scale and intensifies lateral buoyancy gradients, disrupting thermal wind balance and thereby promoting cross-frontal secondary circulations in the frontal zone.”

209 LCS – abbreviation is not presented

Response: We have revised the text to use the full term “Lagrangian Coherent Structure” rather than its abbreviation in Lines 242-243 of the revised version.

212-216 “the scale across the front becomes small enough and the geostrophic balance can no longer be maintained” What do you mean by “small enough”? Less than Rossby radius of deformation? How this decreasing the cross-front scale affects the geostrophic balance?

Response: We thank the reviewer for point this out and have revised the original sentence “As strain fields compress the front, the cross-frontal buoyancy gradient intensifies and the cross-frontal scale decreases, driving frontogenesis and increasing the available potential energy at the front. When the local Rossby number ($Ro = \zeta/f$) approaches $O(1)$, thermal wind balance becomes disrupted, triggering ageostrophic secondary circulations that act to restore this balance” in Lines 250-255 of the revised manuscript.

545-547 Ri characterizes stability of the vertically sheared flow. $Ri \gg 1$ means that Kelvin-Helmholtz instability cannot be generated. I do not see the direct relation between Ri and geostrophic-ageostrophic separation, which is basically linked to the Rossby number.

Response: We appreciate the reviewer’s comment. We clarify here that the Ri used in our study is the geostrophic Richardson number ($Ri = f^2 N^2 / b_x^2$), which is derived from the conventional stability formulation for vertically sheared flows ($Ri = N^2 / (\partial v / \partial z)^2$) through the thermal wind relation. Unlike the traditional Ri , which characterizes vertical shear instability, this geostrophic Ri directly characterizes the disruption of thermal wind balance due to intensified lateral buoyancy gradients. Therefore, low geostrophic Ri values indicate conditions favorable for ageostrophic secondary circulations associated with frontogenesis, closely linked to increased Rossby numbers. We have clarified this distinction in the revised manuscript (Lines 610–622). Additionally, to avoid any confusion, we have consistently used the term “geostrophic Richardson number” throughout the manuscript.

References

1. Ailin Brakstad, Kjetil Våge (University of Bergen, Norway), Sólveig Rósa Ólafsdóttir (Marine and Freshwater Research Institute, Iceland), Emil Jeansson (NORCE Norwegian Research Centre, Norway), and Geoffrey Gebbie (Woods Hole Oceanographic Institution, USA) (2023) Hydrographic and geochemical observations in the Nordic Seas between 1950 and 2019
2. Bashmachnikov, I., Belonenko, T., Kuibin, P., Volkov, D., & Foux, V. Pattern of vertical velocity in the Lofoten vortex (the Norwegian Sea). *Ocean Dyn.* **68**, 1711–1725 (2018).
3. Bashmachnikov, I.I., Boutov, D.D., & Dias, J.J. Manifestation of two meddies in altimetry and sea-surface temperature. *Ocean Sci.* **9**, 249–259 (2013)
4. Bosse A., Fer I., Lilly J. M. & Søiland H. Dynamical controls on the longevity of a non-linear vortex:

The case of the Lofoten Basin Eddy. *Sci. reports* **9**, 13448 (2019).

5. Kohl, A. Generation and stability of a quasi-permanent vortex in the Lofoten Basin. *J. Phys. Oceanogr.* **37**, 2637–2651 (2007).
6. Santeva, E.K., Bashmachnikov, I.L. & Sokolovskiy, M.A. On the Stability of the Lofoten Vortex in the Norwegian Sea. *Oceanology* **61**, 308–318 (2021).

Response to Review Comments for

“Warm Rings in Mesoscale Eddies in a Cold Straining Ocean”

Huizi Dong, Meng Zhou, James C. McWilliams, Roshin P. Raj, Francesco d’Ovidio, Ilker Fer, Lixin Qu, Bo Qiu, Lia Siegelman, Zhengguang Zhang, Walker O. Smith, Jr., Ann Kristin Sperrevik

Note: Reviewer’ comments are in italic font; authors’ response comments are in normal font. Revisions in the revised manuscript are highlighted. Figs. 1–6 are presented in the revised main manuscript; Figs. S1–S9 are included in the Supplementary Information; and Figs. R1–R3 are provided in this response document.

REVIEWER COMMENTS

Reviewer #2 (Remarks to the Author):

As requested by the editor of Nature Communications, I reviewed the manuscript by Dong et al., titled “Warm rigs in mesoscale eddies in a cold straining ocean”. This study provides a thorough analysis of the vertical heat transport associated with submesoscale filaments around mesoscale eddies in the Lofoten Basin of the North Atlantic, and the resulting formation of surface warm ring structures, based on long-term SeaGlider observation data. I believe it is rare to find such a beautiful visualization of heat advection by submesoscale structures from actual ocean observations. Therefore, I support its publication in Nature Communications. However, there are several critical concerns described below. Please clarify these points.

Major concerns:

1. TS Distribution & Salinity Contribution:

- Request to see the TS distribution that determines the density of seawater.*
- Accurate quantification of salinity’s contribution to density and geostrophic flow is needed.*
- Figure 6b implies that temperature is dominant, but in high-latitude regions (70-72°N), salinity typically plays a more significant role in density-driven flows (e.g., Rudels et al. 2013, doi:10.1016/j.pocean.2013.11.006).*

Response: We thank the reviewer for pointing this out. We agree that salinity generally plays a significant role in determining density and associated geostrophic flows in high-latitude regions (Rudels et al., 2013). Our study specifically focused on localized submesoscale processes occurring at eddy edges, where temperature contributions locally surpass salinity variations, thus influencing the local density structure.

Following the reviewer's suggestion, we have included the salinity transect and associated T–S diagrams, and further quantified the relative contributions of temperature and salinity to density and geostrophic flow. The vertical distributions of temperature and salinity along Transect 1 (Fig. R1b–c) illustrate the region's typical hydrographic structure, where temperature variations (2–4°C) surpass salinity variations (<0.1–0.2 psu), influencing the local density significantly.

As illustrated in the T–S diagram (Fig. R1f), below the mixed layer, data points are primarily aligned along the temperature axis and cross multiple isopycnals, indicating the important role of temperature variations on density changes relative to salinity. Further examination of the T–S distribution at 100-m depth intervals (Fig. R1g) made this pattern clearer, supporting the significant influence of temperature on local density gradients.

Using the equation of state and thermal wind balance, we decomposed density gradients into temperature and salinity components and quantified their respective contributions to density (Fig. R1d) and geostrophic flow (Fig. R1e). The results, presented as depth-averaged values with a 40-km moving average, show that temperature generally accounts for over 50% of the contribution—averaging 71.5% for density and 69.9% for geostrophic flow—while salinity contributes only 28.5% and 30.1%, respectively. In the three mesoscale eddy regions identified along the transect (approximately 100–200 km, 300–420 km, and 460–520 km), the temperature contribution increases further, reaching 80.9% for density and 76.8% for geostrophic flow.

Additionally, Fig. S1 presents T–S diagrams from other Seaglider missions, which demonstrate that salinity plays a significant role in density variations within the surface mixed layer. However, at eddy edges where submesoscale processes are most active—the primary focus of our study—temperature gradients strongly dominate density gradients from below the mixed layer down to approximately 500 m depth.

We have included T-S diagrams in Figs. S1-2 and provided analyses of temperature-salinity structure associated with anticyclonic and cyclonic eddies in Lines 182-192 of the revised version.

Fig. R1 | Contributions of temperature and salinity to density and geostrophic flow. Vertical section obtained from Seaglider measurements showing: **a** buoyancy, **b** temperature, and **c** salinity (the same section as Fig. 2 of the main text). **d** Contributions of temperature and salinity to density along the section. **e** Contributions of temperature and salinity to geostrophic flow along the section. **f** T-S distribution across depths from 0 to 600 m along the section. **g** T-S distribution similar to **f** but show only for five depth levels (50, 150, 250, 350, and 450 m).

2. Definition of "Edge" (L92, L98):

- The definition of "edge" is unclear throughout the paper.

- The use of " $r = R$ " to describe the "eddy edge" is questionable. A clear definition of R is needed, possibly the location of maximum azimuthal velocity.

Response: We agree with the reviewer that the definitions of " R ," " R_x ," and "eddy edge" are critical yet unclear in our original manuscript.

Here, " R " represents a unified, normalized radius used in the composite eddy-centric coordinate system for eddies of varying sizes (as in Figs. 3 and S8, which displays features between $0-2R$).

In contrast, " R_x " refers specifically to the radius of individual eddies. As the reviewer mentioned, R_x is defined based on the closed SSH contour associated with the maximum azimuthal velocity.

More explicitly, R_x is the radius of a circle with an area equivalent to that enclosed by the identified SSH contour corresponding to the maximum geostrophic speed. To determine this area accurately, we employed our hybrid eddy detection method, incorporating an Okubo–Weiss (OW) parameter criterion (W) to exclude spurious detections due to noise and ambiguous multipolar or elongated contours, thereby obtaining more robust and consistent eddy boundaries (detailed in Lines 655–675 of the revised manuscript).

Many previous studies^{2–6} on mesoscale dynamic processes have similarly defined the eddy radius (R_x) based on this method. Assuming that the distance between a sampling point and the mesoscale eddy center is d :

- Eddy interior: defined as the region inside the SSH contour ($d/R_x \leq 1$).
- Eddy exterior: defined as the region outside this SSH contour ($d/R_x > 1$).

Given our primary focus on submesoscale dynamics occurring at the eddy edges, we have further delineated three dynamical regions:

- Eddy interior: regions where $d/R_x < 0.8$.
- Eddy edge: regions where $0.8 \leq d/R_x \leq 1.2$.
- Eddy exterior: regions where $d/R_x > 1.2$.

This study we mainly consider the range $d/R_x < 2$ (i.e., the normalized domain within $2R$ as shown in Fig. 3, followed Chelton et al., 2011). It is important to emphasize two key points:

(1) The three-region division described above (interior, edge, and exterior) is used exclusively for descriptive and analytical purposes in the manuscript, and was not utilized in any calculation or data processing step. All Seaglider sampling points, regardless of their position (interior $< 0.8R$, edge $0.8–1.2R$, or exterior $> 1.2R$), were analyzed based solely on their actual positions (d) relative to eddy centers and their angular projections (e.g., relative to true north), as clearly illustrated in Fig. S8.

(2) We selected the radial range of $0.8–1.2R$ for the eddy edges because the majority of mesoscale eddies observed in this study have radii in the range of approximately 30–50 km (Fig. R3). This choice translates into an eddy edge width of approximately 12–20 km, sufficiently broad to encompass and resolve the submesoscale signals (typically < 10 km) that are the main focus of this study.

Although our study did not rely on the three-region definition (interior, edge, and exterior) for any calculations, we fully agree with the reviewer that clear definitions of “ R ,” “ R_x ,” and “eddy edge” are essential. Accordingly, we have reorganized the Methods section of the manuscript (sections “Identification of Mesoscale Eddies” and “Eddy-centric Collocations”) by adding detailed explanations, ensuring consistency throughout the manuscript. In addition, we have expanded and improved the description of the hybrid eddy detection algorithm. For more comprehensive details, please refer to Lines 676–708 in the revised manuscript.

3. Relative Vorticity & Rankine Vortex (L463):

- Request to see the spatial distribution of relative vorticity (ζ) in eddy-centric coordinates.
- Suggest assuming a Rankine vortex profile ($\zeta = \text{constant}$ for inner regions; zero for outer regions) to clarify the positioning of R .

Response: As requested by the reviewer, the spatial distributions of relative vorticity (ζ) in eddy-centric coordinates for ACE and CE are presented in Fig. R2.

We have carefully clarified the definitions and selections of the radii " R " and " R_x " in the revised manuscript and provided detailed explanations in response to the previous comment. Therefore, we have chosen not to employ the Rankine vortex assumption here.

Fig. R2 Eddy-centric composite distributions of relative vorticity (ζ) for **a** anticyclonic eddies and **b** cyclonic eddies in normalized coordinates. Solid and dashed black contours represent positive and negative relative vorticity (ζ), respectively, with contour intervals of $2 \times 10^{-6} \text{ s}^{-1}$. **c** Radial distribution of the magnitude of relative vorticity (ζ) for anticyclonic eddies (red lines) and cyclonic eddies (blue lines). R denotes the normalized radius of the eddy.

4. Geostrophic Velocity Resolution:

- *Questioning whether the geostrophic velocity (u_g, v_g) can be truly resolved from sea surface height (SLA).*

- *Concern about whether the satellite data resolution (0.25×0.25 degrees, ~ 27 km latitudinal resolution) is sufficient to resolve the eddy's horizontal structure, given the deformation radius (~ 10 km).*

Response: We appreciate the reviewer's concerns regarding the resolution of geostrophic velocities derived from satellite altimetry data. In the Lofoten Basin, our previous study (Raj et al.⁷) quantitatively assessed this issue and found that altimeter-derived geostrophic speeds of eddies were approximately 25-30% lower than the geostrophic velocities measured concurrently by surface drifters within the eddies. This discrepancy is attributed to the smoothing effect inherent in the gridding of altimeter data, along with additional influences from centrifugal acceleration and wind-slip on the drifter velocities. These details have been clarified in the revised manuscript (Lines 531-536).

Regarding the resolution limitations for eddies, Chelton et al.² demonstrated that while satellite altimetry data has a grid resolution of $1/4^\circ \times 1/4^\circ$, the effective wavelength resolution of the merged SSH fields is approximately 2° in both latitude and longitude. By fitting a Gaussian SSH structure to the positive half of a cosine with a 2° wavelength, the corresponding eddy e-folding radius (L_e) is determined to be 0.4° , which translates to approximately 15 km at 70° latitude. Consequently, eddies with radii smaller than 15 km are filtered out and cannot be detected in the SSH fields.

To better address the reviewer's concerns, we plotted the frequency distribution of eddy radii in the northern Norwegian Sea ($0-20^\circ\text{E}$, $65-72^\circ\text{N}$) during 2000-2020 (Fig.R3a-b). The results indicate that eddies detected by satellite altimetry in this region have radii ranging from 20-60 km, with peak frequency occurring at 30-40 km. The eddies investigated in this study, specifically those traversed by the Seaglider, have radii between 30-50 km, with anticyclonic eddies of approximately 40 km radius being most numerous. While the local deformation radius is around 10 km, our analysis focuses on these mesoscale eddies with 30–50 km radii that are well resolved by the available altimetry data.

We also compared satellite-derived geostrophic velocities with those obtained from Seaglider measurements, including both geostrophic velocities calculated from glider-measured density fields and depth-averaged currents (DAC) derived from glider trajectories. The comparison revealed that although satellite altimetry data typically underestimate current magnitudes, they can capture the characteristics and spatial structures of mesoscale eddies (figures not shown). Additionally, altimetry data with $0.25^\circ \times 0.25^\circ$ resolution likely underestimates the total number of eddies, particularly those with radii smaller than 20 km, which consequently leads to an underestimation of the submesoscale VHT addressed in this study. Nevertheless, the vertical heat transport induced by mesoscale eddies with radii of 30-60 km, which is the focus of our current research, still carries significant statistical and scientific importance.

All detected eddies 2000-2020

All detected eddies during Seaglider period 2012-2017

Fig. R3 Frequency distribution of **a** anticyclonic eddies, **b** cyclonic eddies of different radii in the northern Norwegian Sea (0-20°E, 65-72°N) between 2000-2020 identified by the hybrid eddy detection algorithm. Frequency distribution of **c** anticyclonic eddies and **d** cyclonic eddies of different radii detected during Seaglider operations (July 2012- August 2017). n and n^* represent the number of anticyclonic and cyclonic eddies, respectively.

Specific comments:

L31-32: having “poleward” appear twice in the same sentence feels repetitive. You should rephrase the sentence to avoid the redundancy. Like “heat loss during its transit towards the polar regions.”

Response: We agree with the reviewer and have revised the sentence as “The warm and saline Atlantic Water has long been recognized as being subjected to a substantial heat loss during its transit towards the polar regions” in Lines 33-34 of the revised manuscript.

L60: Clarify if “stabilized” means the eddy disappears. Consider using “weakened” or “dissipated.”

Response: The term “stabilized” could indeed lead to some confusion. In the Lofoten Basin, the bottom topography plays a crucial role in attracting the anticyclonic eddy into the basin, thereby contributing to the dynamical stability of the LBE. In the revised version (line 64), we have replaced “stabilized” with the more appropriate term “maintained” and added three references to support this clarification (Kohl, 2007; Bosse et al., 2019; Santeva et al., 2021).

L67: Note that heat diffusion around polar eddies also involves mechanisms like double diffusion or mixing by internal waves (e.g., Kawaguchi et al. 2012, doi:10.1016/j.dsr.2012.04.006; Kawaguchi et al. 2016, doi:10.1175/JPO-D-15-0150.1).

Response: We thank the reviewer's comment and have added the sentence "Vertical heat exchange has been reported to be significantly influenced by processes such as double-diffusive convection and internal-wave-driven mixing in polar regions (e.g., Kawaguchi et al., 2012, 2016)" in Lines 69-71 of the revised version. Necessary references have also been included in the revised version.

L71: The origin of the 0.1-10 km range is unclear. The upper bound should relate to the deformation radius.

Response: We have clarified that the 10 km upper bound corresponds approximately to the local Rossby deformation radius in Lines 79-80 of the revised manuscript.

L78: "frontogenesis" should not be capitalized.

Response: The capitalization of "Frontogenesis" has been corrected to "frontogenesis" in Line 86 of the revised manuscript.

L93: A description of the TS structure of cyclones and anticyclones below the mixed layer is necessary. Mention that salinity contributes minimally to density-driven GQ flow.

Response: We have included the necessary content in Lines 184-192 of the revised version, which covers the analysis of temperature and salinity structure of both eddies below the mixed layer, and notes that temperature dominates density variations in this region.

L115: The term FSLE should be defined upon its first mention.

Response: We have defined the term "finite-size Lyapunov exponent (FSLE)" upon its first mention in Lines 123-124 of the revised manuscript.

L123: Provide a simplified definition of Ri or reference the relevant Methods section.

Response: We have provided a definition of Ri and referenced the Methods section at its first occurrence in the revised manuscript (Line 133).

L176: What is the relative contribution of temperature and salinity to restratification?

Response: To clarify our statement, we have revised the original phrase, "the submesoscale secondary circulations act as restratification processes," to "the submesoscale secondary circulations act to flatten sloping isopycnals at the fronts, thus restoring stratification" (Lines 208-210 of the revised manuscript).

A more detailed analysis of the dominant contribution of temperature can be found in our response to the reviewer's Comment 1.

L201: Clarify the "hyperbolic region along the edges." Is it inside or outside?

Response: We thank the reviewer for raising this point, and we have clarified this issue in Line 244-247 of the revised manuscript.

L209: Define "LCS."

Response: We have defined the abbreviation "Lagrangian Coherent Structure (LCS)" upon its first mention in Line 242-243 of the revised version.

L213: The statement "frontogenesis develops by releasing the available potential energy of the front" might be inaccurate. If internal density surfaces stand vertically, wouldn't APE increase?

Response: We have revised the inaccurate statement, and the sentence now reads: "..., the cross-frontal buoyancy gradient intensifies and the cross-frontal scale decreases, driving frontogenesis and increasing the available potential energy at the front" in Lines 251-253 of the revised manuscript.

L216: The "warm side to cold side" assumption only applies if the density profile is temperature-driven. In polar oceans, this could be reversed. Start by explaining the TS structure inside and outside the eddy.

Response: We agree with the reviewer that the "warm side to cold side" assumption requires temperature-dominated density profiles. We've clarified in the text (Lines 182-192) that despite the high latitude (69-72°N), temperature gradients strongly dominate density variations (contributing 71.5% overall and 80.9% in eddy regions) below the mixed layer down to ~500m depth. Further details are provided in our response to the reviewer's first question. Necessary figures have been included in the Figs. S1-2 in the Supplementary Information.

L220: Similar clarification is needed: "Warm fluids are located within the anticyclonic eddy, whereas cold fluids are ..."

Response: Following the reviewer's suggestion, our analysis has shown temperature to be the primary driver of density variations (Lines 184-192). We have therefore revised this sentence to: "Warm, less dense waters occupy the interior of anticyclonic eddies and exterior of cyclonic eddies, while cold, dense waters characterize the interior of cyclonic eddies and exterior of anticyclonic eddies" (Lines 260-262 of the revised manuscript).

L474: The dimensions of "delta_i" and "delta_f" (set at 0.02° and 0.4°) are unclear. Are these distances and assumed as constants? Also please clarify the independent variable in equation (1).

Response: The parameters δ_i and δ_f represent the initial and final separation distances between fluid parcels, expressed in degrees as angular distances (non-Euclidean) on the Earth's surface. These values (0.02° and 0.4°) are indeed constants, characterizing the scales at which we track divergence of neighboring particle trajectories. The Lyapunov exponent $\lambda(x, t, \delta_i, \delta_f) = \frac{1}{\tau} \log\left(\frac{\delta_f}{\delta_i}\right)$ essentially focuses on the relative separation ratio (δ_f/δ_i), which becomes dimensionless after taking the logarithm, regardless of whether the distances are measured in kilometers or angular degrees. The independent variable in Equation (1) is the position vector x , and the Lyapunov exponent λ quantifies the local exponential rate of separation between initially nearby trajectories starting at position x over the integration time τ .

We have revised the paragraph to explicitly define these parameters and clarify the dimensional consistency of the equation (Lines 550-557 of the revised manuscript).

L529: Clarify whether "s is the along-track direction" means distance instead of "direction."

Response: Thank you for pointing this out. "s" refers to the along-track distance. We have corrected this in Line 599 of the revised version.

L542: The use of "N" for buoyancy frequency and "N" for sampling number (Fig. 3) overlaps. Change one of the notations.

Response: In the revised manuscript, all instances of "N" and "N*" in the figures have been replaced with "n" and "n*".

L545: Ri's formula should be presented separately and referenced earlier in the text.

Response: We have presented the R_i separately and referenced earlier in Lines 133-135 of the revised manuscript.

L577: "anomalies" should be clarified—anomalies relative to what?

Response: In the revised version (Lines 650-652), the sentence now reads: “ w' and T are the anomalies of vertical velocity and temperature, respectively, after removing the sectional mean.”

L579: Use consistent notation for $|b_x|$ —is it b_x or B_x ?

Response: We have revised inconsistent notations in both the text and figures to consistently use b_x . We retain $|b_x|$, which represents the absolute value of b_x .

L587: Provide the definition of R_x .

Response: The definitions of “ R ,” “ R_x ,” and “eddy edge” have been comprehensively explained in the main text.

The definitions of “ R_x ” has been comprehensively explained in Lines 658-666 and 680-691 of the revised manuscript.

L590: Clarify "position relative to the eddy" as being either inside or outside relative to the eddy's core.

Response: We have thoroughly revised the description provided in Lines 681-684 of the revised manuscript.

Figures and captions:

Figure 1 (L288, L291, L295):

- Blue dots on the blue background are hard to see; change the color of the background or the dots.
- Provide full forms of SST, CHL, and FSLE for clarity.
- Orange dots on the red background are difficult to see.
- In panel c, showing a vertical section of temperature and salinity from the gliders (even in the supplementary material) would be helpful.

Response: The color representing cyclonic eddies in Fig. 1a has been changed to green dots; the orange dots in Fig. 1c have been replaced with blue dots. Additionally, we have provided the full forms for SST (sea surface temperature), CHL (chlorophyll), and FSLE (finite-size Lyapunov exponent), and included temperature and salinity sections in Fig.S2 of the Supplementary Information.

Figure 2 (L306):

- The notation for "Bx" seems inconsistent. Check whether "Bx," "B_x," "b_x," and "bx" are mixed up.

Response: Thank you for pointing this out. We have checked and revised the notation in the manuscript and figures for consistency.

Figure 3 (L321):

- The panel order in the caption is inconsistent. Either follow alphabetical order or change the panel layout for clarity.

- Consider separating the left and right groups of panels with space and make the variables (Strain, R_i , SSTa) more prominent on the right.

- The definition of "R" as the normalized radius of the eddy is unclear. Is R the point where azimuth velocity becomes maximal?

- The notation for "N" and "N*" in panels a and b should be changed to avoid confusion with buoyancy frequency.

Response: Following the reviewer's suggestion, the caption for Fig. 3 has now been described in alphabetical order. The definition of R has been clarified in our response to Question 1 and explicitly stated in Lines 655-674 and 679-681 of the revised manuscript. Additionally, the symbols "N" and "N*" in panels a and b have been corrected to "n" and "n*", respectively.

References

1. Rudels B, Schauer U, Björk G, et al. Observations of water masses and circulation with focus on the Eurasian Basin of the Arctic Ocean from the 1990s to the late 2000s. *Ocean Sci.* **9**, 147-169. (2013).
2. Chelton, D. B., Schlax, M. G., & Samelson, R. M. Global observations of nonlinear mesoscale eddies. *Prog. Oceanogr.* **91**, 167–216 (2011).
3. Gaube, P., Chelton, D. B., Strutton, P. G., & Behrenfeld, M. J. Satellite observations of chlorophyll, phytoplankton biomass, and Ekman pumping in nonlinear mesoscale eddies, *J. Geophys. Res. Oceans* **118**, 6349–6370 (2013).
4. Dawson, H. R. S., Strutton, P. G., & Gaube, P. The unusual surface chlorophyll signatures of Southern Ocean eddies. *J. Geophys. Res. Oceans* **123**, 6053–6069. (2018).
5. Gaube, P., McGillicuddy Jr., D. J., Chelton, D. B., Behrenfeld, M. J., & Strutton, P. G. Regional variations in the influence of mesoscale eddies on near-surface chlorophyll. *J. Geophys. Res. Oceans* **119**, 8195–8220 (2014).
6. Chelton, D. B., Gaube, P., Schlax, M. G., Early, J. J. & Samelson, R. M. The influence of nonlinear mesoscale eddies on near-surface oceanic chlorophyll. *Science* **334**, 328–333 (2011).
7. Raj, R. P., & Halo, I. Monitoring the mesoscale eddies of the Lofoten Basin: Importance, progress, and challenges. *Int. J. Remote Sens.* **37**, 3712–3728. (2016).

Response to Review Comments for

“Warm Rings in Mesoscale Eddies in a Cold Straining Ocean”

Huizi Dong, Meng Zhou, James C. McWilliams, Roshin P. Raj, Francesco d’Ovidio, Ilker Fer, Lixin Qu, Bo Qiu, Lia Siegelman, Zhengguang Zhang, Walker O. Smith, Jr., Ann Kristin Sperreik

Note: Reviewer’ comments are in italic font; authors’ response comments are in normal font. Revisions in the revised manuscript are highlighted. Figs. 1–6 are presented in the revised main manuscript; Figs. S1–S9 are included in the Supplementary Information; and Figs. R1–R2 are provided in this response document.

REVIEWER COMMENTS

Reviewer #3 (Remarks to the Author):

1) The study uses glider observations near eddies to investigate vertical heat transport (VHT) driven by submesoscale processes down to 400 meters, with claims that this deep-reaching VHT warms the upper ocean and contributes to warm eddy rings. While the topic is promising, the manuscript currently suffers from significant issues in methodology, theoretical foundation, and dynamical interpretation. The authors seem to have limited experience with glider observations and an incomplete understanding of geophysical fluid dynamics, especially concerning high-frequency, small-scale ocean processes. Given these gaps, I believe that substantial portions of the study should be restructured, potentially with greater reliance on model results for a more reliable analysis.

Response: We thank the reviewer for the comprehensive comments, which we have addressed in detail below and which have considerably helped improve the manuscript.

2) In its first part, the study largely repeats findings from Siegelman et al. (2020), employing similar data but in a different region, with fewer mechanisms discussed. Scientific innovation diminishes when a concept is reiterated without substantial new insights; hence, this section lacks impact.

Response: We thank the reviewer’s concern regarding overlap with Siegelman et al. (2020). Our work makes several distinct contributions:

- This is the first detailed submesoscale dynamics study in the Lofoten Basin, a critical Atlantic Water transformation zone (Lines 53-57, 64-67).
- Finding similar mechanisms in a completely different oceanic region (sub-Arctic vs Antarctic) provides important validation of these processes' universality.
- Our five-year observational dataset (versus Siegelman's three months) significantly enhances statistical robustness and reveals these processes' consistent presence to ~500m depth.

- We address a specific knowledge gap regarding submesoscale contributions to vertical heat transport in this eddy-rich region, which has implications for Atlantic Water transformation.
- Based on the reviewer's insightful comment, we have included an analysis of mesoscale VHT. Thus, our study represents the first quantitative comparison between submesoscale and mesoscale vertical velocities and their contributions to VHT in the Lofoten Basin region.

These aspects collectively represent a meaningful advancement rather than simply reiterating previous findings.

3) The second part includes a composite analysis describing “warm rings” within eddies, a feature well-documented in previous literature. Although the authors attribute these rings to submesoscale VHT, there is no direct evidence linking the rings specifically to submesoscale rather than mesoscale dynamics, making the analysis questionable.

Response: We appreciate the reviewers’ comments on this point. To demonstrate that ‘warm-rings’ within eddies are predominantly driven by submesoscale rather than mesoscale dynamics, we conducted additional analyses comparing VHT contributions from both processes (described in detail in the new Supplementary Information section "Comparison between mesoscale and submesoscale VHT", Lines 148-177).

Although we acknowledge that submesoscale motions are not the only mechanism contributing to the formation of warm ring structures, our analyses reveal clear differences in both intensity and spatial distribution between submesoscale and mesoscale vertical processes:

- Submesoscale vertical velocities (~60 m/day) substantially exceed mesoscale vertical velocities (<10 m/day) by nearly an order of magnitude. Similarly, submesoscale VHT (up to 1400 W/m²) is considerably stronger than mesoscale transport (Fig. S5).
- Spatially, submesoscale processes are predominantly concentrated at eddy edges, creating the observed "warm-ring" structures, whereas mesoscale processes are distributed more broadly across both eddy edges and interiors, typically forming classical “warm-core” or “cold-core” structures (Figs.S5-7, and Gauble et al., 2013, 2014; McGillicuddy, 2016).

In addition to the mesoscale vertical motions mentioned above, other mechanisms related to warm-ring structures, such as mesoscale eddy stirring, have also been well documented in previous literature²⁻⁵. Since the eddy stirring mechanism primarily involves horizontal transport, whereas submesoscale processes are characterized by intense vertical motions that promote vertical exchange of materials and heat, these two processes can be clearly distinguished:

To further substantiate the crucial role of submesoscale vertical motions at eddy edges, we present an additional case study based on Seaglider data from May 2017 (Fig. R1). This section was specifically chosen because the Seaglider crossed the edge of the same anticyclonic eddy three times during the one-month period (Fig. R1a,b). Corresponding nutrient and chlorophyll profile

observations (Fig. R1g–i), obtained from the NMDC database⁶ (<https://doi.org/10.21335/NMDC-1271328906>) during the same period, allowed us to analyze vertical transport patterns at both eddy-edge and non-edge regions.

The observations revealed pronounced lateral buoyancy gradients and strong vertical velocities at the eddy edges (Fig. R1c–f), which facilitated enhanced upward transport of nutrients (e.g., nitrate, silicate) into the surface layer (Fig. R1g–i). The increased surface nutrient concentrations subsequently led to elevated chlorophyll-a concentrations, providing indirect but compelling evidence of intensified submesoscale-driven vertical exchange at the eddy edges (Figs. R1g–i).

Similar results from additional cases involving both anticyclonic and cyclonic eddies (Fig. R2) consistently show that submesoscale vertical circulation at eddy edges significantly enhances nutrient availability and productivity relative to eddy interiors and external waters. This body of evidence collectively demonstrates that submesoscale-driven vertical transport mechanisms play a dominant role in the formation of warm-ring structures at eddy edges.

4) A critical flaw of the method is the observing platforms. In the past two decades, observations using gliders do improve the observation skills especially about meso- and smaller-scale physics. Gliders may observe mesoscale features, but gliders, with a slow movement of approximately 10 km/day, cannot adequately resolve lateral submesoscale features. As established by Rudnick and Cole (2011, doi:10.1029/2010JC006849), the Doppler smearing and aliasing effects in glider data lead to significant errors in the submesoscale range, particularly in stratified layers. I think the author need to carefully go through this paper before working on this study. In fact, Studies typically use gliders for vertical profiles or along-isopycnal analyses, or in networked arrays to address resolution issues (e.g., Rudnick et al. 2022, doi:10.1175/JPO-D-21-0181.1). Consequently, I find the buoyancy gradients presented in Fig. 1e unreliable, which casts doubt on the Richardson number (Ri) and vertical velocity (w) estimates. The results of VHT are quite questionable unless the authors can demonstrate that the smearing and aliasing effects have limited impacts. As suggested in the recent several field campaigns these years, observations using onboard towed bodies (with ship speed >6 knots) are preferable to minimize such errors, although this is challenging.

Response: We agree with the reviewer's concerns that the sampling using a Seaglider may limit the resolution of lateral submesoscale features, particularly due to the introduction of Doppler smearing and aliasing effects, potentially impacting the accuracy of our results significantly (Rudnick et al.^{7,8}). To address this challenge, we invited Ilker Fer (University of Bergen), the original principal investigator of these seaglider observations, to join our work. Fer has experience in glider data processing and interpretation of observations obtained from gliders. During discussions with Fer, and in response to key issues raised by the reviewer, we recognized that the near-real-time CMEMS dataset initially utilized were not optimal for our analysis. We have therefore replaced the CMEMS

dataset with the fully quality-controlled and calibrated Seaglider dataset, which is archived at the Norwegian Marine Data Centre (NMDC) under a CC BY 4.0 license^{9,10}.

All analyses presented in the revised manuscript have been fully reprocessed using this updated NMDC Seaglider dataset. New analysis includes substantial smoothing and objective mapping. Additionally, we have included a new section of “Analysis of the Seaglider data” in the Supplementary Information.

The calibrated and processed results for Transect 1 are shown in Fig. 2 (2 km bin averaged profiles are unsmoothed; lateral gradients are smoothed over 10 km to reduce noise) and Fig. S9 (objectively mapped fields). Although objective mapping was applied to mitigate Doppler smearing and aliasing effects, strong lateral buoyancy gradients and vertical velocity signals are still clearly identifiable at the mesoscale eddy edges. This suggests that the submesoscale features we observed have likely been reasonably well captured, supported by the following considerations:

- (1) The magnitude of observed submesoscale motions substantially exceed potential scale-mixing and signal distortions caused by Doppler smearing and aliasing effects, suggesting these effects have limited impact. These effects are minimal, as evidenced by the clear presence of submesoscale signals that remain identifiable even after applying the 10 km smoothing filter.
- (2) The pronounced lateral buoyancy gradients (b_x) and vertical velocities (w) observed at mesoscale eddy edges exhibit spatial structures and magnitudes that are physically consistent. These features align well with established submesoscale dynamics and are quantitatively consistent with values reported in the literature (Klein & Lapeyre, 2009; Siegelman et al., 2020).
- (3) A statistical analysis across all glider missions (M1–M8) indicates that vertical velocities and vertical heat transport are typically underestimated by approximately 15–20% after applying a 10 km Gaussian smoothing filter.
- (4) In line with the reviewer's suggestions, we have strengthened the modeling analyses in the study. Additional model validation results are included in the ‘Numerical simulation verification’ section of the revised Supplementary Information and the revised manuscript (Lines 149-158, 173-179, 213-216, 302-316, and Figs. S3–4, S7). These comparisons with model results further demonstrate the reliability of the strong buoyancy gradients and VHT observed by the Seagliders at the eddy edges.

5) What do you mean by 3D view of vertical velocity (Line 161 and Fig. 4). The results are just presented without any clear interpretation. The results lack clear interpretation, and it's unclear if they include submesoscale effects. If they do, this would be unexpected. If derived from composite fields, these vertical velocity estimates are likely inaccurate. Using composite eddy fields to infer ageostrophic processes is misleading, as this approach fails to capture true dynamics. It's important to note that the composite field approach is essentially a method to illustrate the general features of the eddy and should not be used to quantify submesoscale characteristics. Composite fields average over time and space, which tends to smooth out smaller-scale variations and can

misrepresent the true intensity and structure of submesoscale processes. Therefore, using composite fields to infer submesoscale dynamics is not methodologically sound and likely introduces significant inaccuracies.

Response: The submesoscale motions and their associated heat transport discussed in this manuscript are derived from individual Seaglider transect measurements, whereas the composite eddy fields serve as a further analysis and visualization of these numerous transect results. To clearly demonstrate to the reviewer that our composite analysis effectively captures submesoscale signals at eddy edges, even under averaged conditions, we take Transect 1 as an illustrative example to comparatively analyze the compositing processes of the submesoscale and mesoscale components (Figs. S5-6):

Taking the mesoscale VHT (VHT_{sub}) as an example (Fig. S5f), we observed pronounced heat transport signals at eddy edges—around distances of approximately 110 km, 400 km, and 500 km along the transect—with values exceeding 1000 W m^{-2} . In the eddy-centric normalized coordinate system (Fig. S6c), these strong heat transport values are projected onto positions close to one eddy radius ($r \approx R$). In contrast, relatively low heat transport values were observed in non-edge regions along the transect, including both the eddy interior and exterior, and these lower values are projected onto positions either inside or outside the normalized eddy radius. Thus, even in averaged conditions, the composite VHT_{sub} field clearly exhibits enhanced heat transport at the eddy edges, forming distinctive "warm-ring" structures (Fig. S6c).

Similarly, for the submesoscale VHT (VHT_{meso}) (Fig. S5g)—even without considering the net transport—relatively strong heat transport signals ($\sim 100 \text{ W m}^{-2}$) are still observed at the eddy edges and within the eddy interior, which are projected onto regions within the normalized radius ($r \leq R$). In contrast, the weaker VHT_{meso} values outside the eddy are projected onto regions beyond the normalized radius ($r > R$). Consequently, the mesoscale composite fields ($|VHT_{meso}|$, Fig. S6d) display intensified heat transport within the eddy, forming a "warm-core" or "cold-core" structure.

By collocating Seaglider sampling data with the nearest eddy and compositing them onto a normalized, eddy-centric coordinate system, we are able to present an integrated view while preserving the fundamental submesoscale characteristics captured in the individual measurements. Fig. 3 represents averaged composite fields of the upper 500m, whereas Fig. 4 provides a three-dimensional view without depth averaging, illustrating distinct layers at depths of 0m, 200m, and 400m. Following the reviewer's suggestion, we have added a section titled "Mesoscale and submesoscale VHT comparison" in the Supplementary Information.

6) *Additionally, the role of mesoscale contributions to VHT is underrepresented. Mesoscale processes likely contribute substantially to VHT, especially in an averaged context, and should be acknowledged.*

Response: In the revised Supplementary Information, we have added a new section “Comparison between mesoscale and submesoscale VHT”. Additionally, in the section “Numerical simulation verification” (see Supplementary Information), we have included a composite analysis of the 2017-2019 model results, comparing submesoscale and mesoscale VHT within the study region.

Our analysis reveals that strong VHT_{sub} is primarily found at eddy edges, whereas VHT_{meso} is distributed across both eddy edges and interiors, with magnitudes approximately an order of magnitude lower. Even without accounting for net heat fluxes, $|VHT_{meso}|$ constitutes only about 1/6 to 1/5 of VHT_{sub} . The dominance of submesoscale-induced VHT is consistent with recent studies:

(1) Su et al. (2020), using LLC4320 numerical simulations, found that submesoscale processes contribute 4-5 times more VHT annually than mesoscale processes.

(2) Cao et al. (2024), based on in-situ high-resolution observations, report that submesoscale- VHT exceeds mesoscale contributions by more than fivefold.

(3) Torres et al. (2025) similarly emphasize that submesoscale dynamic processes (<15 km) contribute over 80% of VHT, far exceeding mesoscale contributions.

We have also incorporated the necessary analyses in the revised manuscript (Lines 149-158 and 303-317).

7) *More calibration details on the glider data are needed, especially given the extended observation period.*

Response: We agree with the reviewer’s comment and have added a section “Quality control and calibration of Seaglider data” in the Supplementary Information.

8) *The sentence on Line 157, “When the scale across the front is compressed to a critical point, geostrophic balance is no longer maintained. By breaking the geostrophic balance that suppresses vertical motions, the fronts support vertical velocities of up to 60 m day⁻¹ below the surface mixed layer,” is confusing. The breakdown of geostrophic balance typically increases vertical motions via non-geostrophic terms, rather than suppressing them.*

Response: We have revised these sentences for clarity as: “As the cross-front scale is compressed by the strain field, frontogenesis intensifies, increasing the buoyancy gradient and ultimately disrupting the thermal wind balance. To restore this balance, an ageostrophic secondary circulation develops, characterized by surface convergence with downwelling on the dense side and upwelling on the lighter side of the front. This secondary circulation generates vertical velocities reaching up to 60 m day⁻¹ below the surface mixed layer, nearly an order of magnitude greater than those associated with mesoscale motions” in Lines 171-178 of the revised version.

9) In summary, the findings presented here are largely redundant with existing literature or are based on analysis and interpretation that lack sufficient rigor. The data-supported results largely replicate previously reported findings, while the remaining conclusions appear to stem from a lack of careful interpretation and logical coherence. This leaves an impression of incomplete novelty and unsound analytical approaches within the study.

Response: We thank the reviewer for their overall assessment of our study. We have now carefully addressed the critical points raised by the reviewer, as detailed above. As a results, the novelty of our study is better emphasized and the robustness of our findings is strengthened.

Fig. R1 | Characteristics of submesoscale motions of anticyclonic eddy edges and their impacts on vertical nutrient distribution. **a** SLA and **b** FSLE fields overlaid with the Seaglider track from 7 May 2017 to 2 June 2017. The Seaglider traveled approximately 450 km from inside the anticyclonic eddy (point S1 in panel **a**) to outside the eddy (point S2), with measurements shown in panels **c-g**. The glider's vertical sections show **c** buoyancy, **d** lateral buoyancy gradient (B_x), **e** vertical velocity (w) and **f** Richardson number (Ri). The mixed layer depth (MLD) is shown by a thick black line in **c-f**. Yellow and green stars in panels **a** and **b** represent BGC stations at the eddy edge and non-edge locations, respectively, on 22 May 2017. **g-i** Vertical profiles of nitrate, silicate and chlorophyll-a (CHL-a) concentrations, where red lines represent profiles at the eddy edge station (yellow star in panel **a**) and black lines show profiles at the non-edge station (green star). The Seaglider crossed the edge of the same anticyclonic eddy three times during the one-month period (indicated by red, blue, and green points in panel **b** and corresponding colored markers on the along-track time coordinate below the panel **f**).

Fig. R2 | Vertical distribution of nutrients and chlorophyll in edge and non-edge regions of two anticyclonic eddies and one cyclonic eddy. **a-c** FSLE and SLA fields for three mesoscale eddy cases on a 22-May-2017, b 13-June-2022, and c 23-July-1995, overlaid with BGC stations at the eddy edge (yellow stars) and non-edge locations (green stars). **d-f** Vertical profiles of nitrate, silicate and chlorophyll-a (CHL-a) concentrations, where red and blue lines represent profiles at the eddy edge station (yellow stars in panels **a-c**) and black lines show profiles at the non-edge station (green stars in panels **a-c**).

References

1. Siegelman, L. et al. Enhanced upward heat transport at deep submesoscale ocean fronts. *Nat. Geosci.* **13**, 50–55 (2020).
2. Gaube, P., McGillicuddy Jr., D. J., Chelton, D. B., Behrenfeld, M. J., & Strutton, P. G. Regional variations in the influence of mesoscale eddies on near-surface chlorophyll. *J. Geophys. Res. Oceans* **119**, 8195–8220 (2014).
3. Gaube, P., Chelton, D. B., Samelson, R. M., Schlax, M. G., O’Neill, L.W. Satellite observations of mesoscale eddy-induced Ekman pumping. *J. Phys. Oceanogr.* **45**:104–32 (2015)
4. McGillicuddy Jr, D. J. Mechanisms of physical-biological-biogeochemical interaction at the oceanic mesoscale. *Annu. Rev. Mar. Sci.* **8**, 125–159 (2016).
5. Siegel, D. A., Peterson, P., McGillicuddy, D. J. Jr., Maritorena, S., & Nelson, N. B. Bio-optical footprints created by mesoscale eddies in the Sargasso Sea. *Geophys. Res. Lett.* **38**, L13608 (2011).
6. Ailin Brakstad, Kjetil Våge (University of Bergen, Norway), Sólveig Rósa Ólafsdóttir (Marine and Freshwater Research Institute, Iceland), Emil Jeansson (NORCE Norwegian Research Centre, Norway), and Geoffrey Gebbie (Woods Hole Oceanographic Institution, USA) (2023) Hydrographic and geochemical observations in the Nordic Seas between 1950 and 2019
7. Rudnick, D. L., & Cole, S. T. On sampling the ocean using underwater gliders. *J. Geophys. Res.* **116**, C08010 (2011).
8. Rudnick, D. L. Ocean research enabled by underwater gliders. *Annu. Rev. Mar. Sci.* **8**, 519–541 (2016).
9. Fer, I. & Bosse, A. Seaglider missions in the Lofoten Basin of the Norwegian Sea, 2012-2015 (2017). <https://doi.org/10.21335/NMDC-UIB.2017-0001>
10. Bosse, A. & Fer, I. Seaglider missions in the Norwegian Sea during the PROVULO project (2019). <https://doi.org/10.21335/NMDC-980686647>
11. Su, Z., Wang, J., Klein, P., Thompson, A. F., & Menemenlis, D. Ocean submesoscales as a key component of the global heat budget. *Nat. Commun.* **9**, 775 (2018).
12. Cao, H., Freilich, M., Song, X., Jing, Z., Fox-Kemper, B., Qiu, B., et al. Isopycnal submesoscale stirring crucially sustaining subsurface chlorophyll maximum in ocean cyclonic eddies. *Geophys. Res. Lett.* **51**, e2023GL105793 (2024).
13. Torres, H. S., Wineteer, A., Rodriguez, E., Klein, P., Thompson, A. F., Perkovic-Martin, D., et al. Submesoscale eddy contribution to ocean vertical heat flux diagnosed from airborne observations. *Geophys. Res. Lett.* **52**, e2024GL112278 (2025).

“Warm Rings in Mesoscale Eddies in a Cold Straining Ocean”

Huizi Dong, Meng Zhou, James C. McWilliams, Roshin P. Raj, Francesco d’Ovidio, Ilker Fer, Lixin Qu, Bo Qiu, Lia Siegelman, Zhengguang Zhang, Walker O. Smith, Jr., Ann Kristin Sperrevik

Note: Reviewer’ comments are in italic font; authors’ response comments are in normal font.
Revisions in the revised manuscript are highlighted.

REVIEWER COMMENTS

Reviewer #2 (Remarks to the Author):

Please refer to the separately uploaded letter for the comments.

Second Review of 'Warm Rings in Mesoscale Eddies in a Cold Straining Ocean' by Huizi Dong et al.

At the request of Nature Communications, I have conducted a second review of the revised manuscript by Dong et al. I would like to first express my appreciation for the authors’ thorough and thoughtful responses to my initial comments. Overall, I am satisfied with the revisions, and I do not have any major concerns that would warrant substantial additional changes. However, I would like to offer the following two comments for the authors’ consideration:

1. TS Distribution & Salinity Contribution

I appreciate the authors’ inclusion of salinity transects and T–S diagrams, as well as the quantitative assessment of the respective contributions of temperature and salinity to density and geostrophic flow.

The vertical distributions of temperature and salinity along Transect 1 (Fig. R1b–c) illustrate the typical hydrographic structure of the region, where temperature variations (2–4 °C) are larger than salinity variations (<0.1–0.2 psu), and thus have a greater influence on local density.

I agree with the authors that, within the parameter range considered (temperature = 4–8 °C, salinity = 35.2–35.4), temperature variations predominantly govern the density structure.

As an additional suggestion, rather than qualitatively describing this point with multiple phrases, I recommend introducing the density ratio, $R_p = (\alpha \Delta T) / (\beta \Delta S)$, which offers a concise and quantitative way to demonstrate the dominant role of temperature anomalies.

Using typical values:

$-\alpha \approx 2 \times 10^{-4} \text{ }^\circ\text{C}^{-1}$

$-\beta \approx 8 \times 10^{-4} \text{ psu}^{-1}$

*the density ratio is approximately $R_\rho \sim 5$, clearly indicating that temperature changes have a*
*much greater impact on density than salinity variations under these conditions. This would help*
*clarify the physical basis of the authors' interpretation and improve the overall readability of the*
*manuscript.*

**Response:** We thank the reviewer for this clear and helpful suggestion. We have now incorporated
the density ratio (R_ρ) into the Results section of the revised manuscript (Lines 186-194) to provide
a quantitative assessment of the dominant role of temperature in controlling density variations.
This addition has indeed improved the clarity and quantitative basis of the manuscript.

*2. Geostrophic Velocity Resolution*

*Thank you for the authors' explanation and response regarding the concern about the resolution*
*of satellite SSH data in detecting mesoscale eddies. I particularly appreciate the quantitative*
*assessment of the typical size of these eddies.*

*While the local deformation radius is around 10 km, our analysis focuses on these mesoscale*
*eddies with 30–50 km radii that are well resolved by the available altimetry data.*

*However, I believe the logic provided in the response is flawed. The key issue is not how many*
*local deformation radii fit into an eddy of 30–50 km radius, but rather whether the satellite SSH*
*data have sufficient resolution to adequately resolve such features.*

*At 70°N, the meridional resolution of a 1/4° gridded product corresponds to approximately 27.8*
*km, meaning that an eddy may only be represented by one or two grid points in the meridional*
*direction—clearly limiting the accuracy of structure detection.*

*On the other hand, the zonal resolution is finer, approximately 9.5 km, allowing for about 3 to 5*
*grid points across the eddy in the zonal direction. This suggests that, at least in the zonal direction,*
*the horizontal velocity structure of the eddy can be reasonably resolved.*

*Assuming the eddies are roughly circular in shape (even if somewhat deformed), I recommend*
*that the authors briefly clarify in the manuscript that resolving eddy structure in the zonal*
*direction may be sufficient to enable satellite-based eddy detection in this region.*

**Response:** We appreciate the reviewer's insightful suggestions regarding satellite SSH data
resolution. Following the reviewer's suggestion, we have added the contents to Methods section of
the revised manuscript (Lines 543-546) accordingly:

*“At our study latitude (~70°N), this gridded product yields a meridional resolution of ~27.8 km*
*and a zonal resolution of ~9.5 km. For the mesoscale eddies with radii of 30-50 km that we focus*
*on, the zonal resolution is sufficient to resolve their horizontal velocity structure given their*
*quasi-circular nature.”*

“Warm Rings in Mesoscale Eddies in a Cold Straining Ocean”

Huizi Dong, Meng Zhou, James C. McWilliams, Roshin P. Raj, Francesco d’Ovidio, Ilker Fer, Lixin Qu, Bo Qiu, Lia Siegelman, Zhengguang Zhang, Walker O. Smith, Jr., Ann Kristin Sperrevik

Note: Reviewer’ comments are in italic font; authors’ response comments are in normal font. Revisions in the revised manuscript are highlighted.

REVIEWER COMMENTS

Reviewer #3 (Remarks to the Author):

I appreciate the substantial revisions made in this version of the manuscript. The authors have addressed some of my previous concerns; however, several major issues remain unresolved. These concerns are critical and may introduce significant uncertainties in the results presented.

1. On the novelty of the findings. While the authors continue to emphasize a connection between warm rings and mesoscale eddies, the core of the study essentially reflects enhanced vertical heat transport at oceanic fronts, as comprehensively addressed in Siegelman et al. (2020) and Cao et al. (2024) where similar dynamics and methodologies are presented. By the way, I think wrong citation for Cao et al. (2024) is listed in the reference. Upon revisiting these two references, I find that both the underlying dynamics and diagnostic methods have already been thoroughly explained in the existing literature. The current manuscript does not appear to introduce substantial new insights beyond these established findings. Although the authors attempt to highlight the elevated values of vertical heat transport (VHT), this alone does not convince me as a significant novelty, especially considering that these values may be overestimated, as discussed below.

Response: We sincerely thank the reviewer for highlighting the connections between our work and the studies of Siegelman et al. (2020) and Cao et al. (2024).

We agree that Vertical Heat Transport (VHT) at oceanic fronts is an important physical phenomenon common to these studies. However, we would like to take this opportunity to articulate more clearly the essential distinctions of our study in terms of its dataset, methodology, and novel scientific findings:

(1) In terms of scientific question, our study is the first to elucidate the role of the submesoscale
on VHT in the Lofoten region with a multi-year in situ dataset. We believe that this is an
important advancement because VHT in this specific area is known to be a key mechanism of
North Atlantic water mass transformation but until now the role of the submesoscales on this
mechanism had remained an open question. The answer to this regional question is clearly not
in Siegelman et al. (2020) or Cao et al. (2024), although these works suggested us some
possible scenarios and provided some methodological tools for exploring our dataset.

(2) We utilize a five-year continuous Seaglider dataset, compared to the three-month mammal
data in Siegelman et al. (2020) and to the limited number of snapshot-like transects of Cao et
al. (2024). This is not merely an increase in data volume; our extensive dataset allows us to
derive more representative conclusions. Notably, our three case studies presented in the
revised manuscript are from July-August, September, and January-February (Figs. 2, S3, S4),
demonstrating the year-round nature of these processes, rather than seasonal snapshots, hence
providing a strong case for their role on water mass transformation.

(3) Our results show that submesoscale VHT signals penetrate to depths of up to 500 m,
extending previous observations limited to the upper 400 m (Siegelman et al., 2020) and 200
52 m (Cao et al., 2024).

(4) Our study specifically focuses the dynamics at the eddy edge. Thanks to the long temporal
coverage of our dataset, we were able to composite and analyze cyclones and anticyclones
separately. This revealed that submesoscale VHT is consistently upward at the edges of both
eddy polarities, providing a dynamical explanation for the "warm ring" structures observed in
satellite SST. These aspects were not addressed in either Siegelman et al. (2020) or Cao et al.
(2024).

Finally, we thank the reviewer for pointing out the citation issue with Cao et al. (2024), which we
have now corrected in the revised manuscript.

*2. Concerns regarding data contamination. The issue of potential data contamination remains*
*insufficiently addressed. While the authors have now described stricter data quality control*
*procedures and implemented a 10-km smoothing filter to mitigate aliasing and smearing effects,*
*there is no clear evidence demonstrating the effectiveness of this approach. A straightforward way*
*to validate the impact of filtering may be to compare the wavenumber spectra before and after*
*applying the filter. Based on my own experience with glider data, this method is unlikely to fully*
*resolve contamination at submesoscales. It appears the authors may have misunderstood the*
*findings of Rudnick et al. (2011): the primary limitation arises from the slow pace of gliders, not*
*the quality of the sensors. Temporal and spatial glider motion can cause substantial aliasing from*
*internal waves, limiting the reliable resolution of submesoscale variability. While gliders are*

*valuable tools for oceanographic observations, their use in submesoscale studies carries inherent*
*challenges that must be more carefully considered here.*

**Response:** We thank the reviewer for the comprehensive comments on the issue of data
contamination. Following the reviewer's comments, we compared the horizontal wavenumber
spectra before and after filtering using the Seaglider transect T1 (shown in Fig. 2 of the main text
as an example). This transect T1 spanned from 6 July 2012 to 16 August 2012, covering
approximately 650 km. We obtained the power spectral density by detrending the interpolated data,
applying a Gaussian window function, performing FFT, and taking the single-sided spectrum. Figs.
R1a-c show the wavenumber spectra of isopycnal depths, while Figs. R1d-f present the
wavenumber spectra of temperature on constant depth surfaces. The results demonstrate that the
10-km spatial filtering significantly reduces the energy of high-frequency components
(wavenumbers > 0.1 cpkm), and submesoscale signals at eddy edges remain observable after
filtering (Fig. 2).

Although our analysis employs objectively mapped and spatially smoothed data, fully eliminating
Doppler aliasing and smearing- and quantifying its contamination- remains challenging. To
address this concern, and as suggested by the reviewer in the last round, we performed additional
along-isopycnal analysis (Fig. R2). Specifically, Fig. R2d-f shows these along-isopycnal analysis
results for transect T1. For ease of comparison, Fig. R2a-c presents corresponding z-coordinate
results using our 10-km filtered data. The along-isopycnal processing involved these steps:

- (1) Projection of the Seaglider's temperature and salinity data from depth coordinates, $T(x, z)$ and
$S(x, z)$, onto isopycnal surfaces, yielding $T(x, \sigma_k)$ and $S(x, \sigma_k)$.
- (2) The depth series of isopycnals, $z(x, \sigma_k)$, were spatially smoothed using a 10-km Gaussian
low-pass filter to obtain dynamically smoother isopycnal depths, $z^*(x, \sigma_k)$, thereby mitigating
undulations caused by internal wave heave. The T/S data on isopycnal surfaces were
smoothed using the same filter. Figure R2d shows the resulting temperature field on isopycnal
surfaces (following Rudnick et al., 2011).
- (3) These smoothed isopycnal depths and T/S data were then interpolated back to the
depth-coordinate grid to reconstruct $T^*(x, z)$, $S^*(x, z)$ fields that are relatively less effected by
aliasing of internal wave signals.

Figs. R2e,f show the lateral buoyancy gradient (bx) and vertical velocity (w) fields for transect T1
calculated from the isopycnally-processed and reconstructed $T^*(x, z)$ and $S^*(x, z)$ fields. Due to
the use of smoothed isopycnal depths $z^*(x, \sigma_k)$ in the reconstruction, the influence of internal wave
heave on these derived fields is substantially reduced. The strong lateral buoyancy gradient and
significant vertical velocities are still prominently observed at the eddy edges. These
isopycnally-derived features agree well in location and general structure with those from our
10-km z-coordinates filtered data (Fig. R2a-c). This agreement suggests that our 10-km

z-coordinate filter effectively captures primary submesoscale structures, mitigates significant
 noise, and yields results comparable to isopycnal processing for these features.

Furthermore, Fig. R3a-c presents wavenumber spectra of potential temperature before and after
 our along-isopycnal processing (with results re-projected to depth coordinates). For direct
 comparison, Figure R3d-f displays the spectra before and after the 10-km Gaussian spatial filter
 was applied directly in depth-coordinates.

We agree with the reviewer that while gliders are powerful tools for oceanic observations, their
 limitations must be carefully considered when studying submesoscales. Accordingly, we have
 expanded the Discussion section (Lines 304-310) and added a new section, ‘Validation using
 along-isopycnal analysis’, to the Supplementary Information (Lines 138-156).

 **Fig. R1 | Wavenumber spectra from the Seaglider transect (6 July 2012 to 16 August 2012)**
 **shown in main text Fig. 2. a-c** Depth spectra for the 27.6, 27.7, and 27.8 kg m⁻³ isopycnal
 surfaces. **d-f** Potential temperature spectra at depths of 100 m, 200 m, and 400 m. In all panels, the
 black and red lines depict the spectra before and after applying a 10-km Gaussian filter,
 respectively. The dashed gray line indicates a k^{-2} slope for reference. Wavenumber is in cycles per
 kilometer (cpkm).

**Fig. R2** | **a** Buoyancy, **b** lateral buoyancy gradient (b_x), and **c** vertical velocity (w) in
 depth-distance coordinates (same as main text Figs. 2c, 2e, and 2g). **d** Potential temperature on
 selected isopycnal surfaces from the Seaglider transect, with isopycnal intervals of 0.2 kg m^{-3} ;
 x -axis shows Seaglider along-track distance. **e** Lateral buoyancy gradient (b_x) and **f** vertical
 velocity (w) calculated in depth coordinates using isopycnal-filtered temperature and salinity data.

**Fig. R3** | a–c Horizontal wavenumber spectra of potential temperature at nominal depths of 100 m
 (a), 200 m (b), and 400 m (c). Black lines show the spectra computed directly from data on depth
 surfaces. Red lines show the spectra for data processed using the along-isopycnal analysis (as
 detailed in our Response to Reviewer Question 2), and subsequently re-projected to the respective
 nominal depth. d–f are identical to Fig. 1 d–f, showing spectra before (black lines) and after (red
 lines) the application of a 10 km Gaussian spatial filter directly in depth coordinates. The dashed
 gray line indicates a k^{-2} slope for reference.

*3. Concerns about composite analysis methodology. I continue to believe that the composite*
 *analysis employed is misleading. The mesoscale eddy structures included are highly variable, and*
 *their differences cannot be adequately captured by composite means. Critically, the strength and*
 *structure of fronts appear to be the primary drivers of VHT, as shown in Siegelman et al. (2020).*
 *These frontal characteristics are in turn influenced by external forcing, background flow, and*
 *possibly eddy–eddy interactions. The manuscript does not sufficiently define or justify how*
 *azimuthal variability is treated or interpreted across the ensemble of eddy cases. Given this*
 *heterogeneity, I strongly recommend that the authors instead present representative vertical*

*sections from selected cases, which would offer a more physically meaningful and interpretable*
*analysis.*

**Response:** We sincerely thank the reviewer for the insightful comments. We agree that given the
heterogeneity of mesoscale eddy structures, the application of composite analysis must be
approached with great care and robust justification.

Following the reviewer's suggestion, we have incorporated two new representative Seaglider
transect cases into the manuscript (Figs. S3 and S4, also shown as Figs. R4 and R5 below), in
addition to the 2017 case discussed in the last-round response. Fig. S3 presents a cross-basin
transect that sampled four anticyclonic and two cyclonic eddies. Fig. S4 shows another transect
that crossed multiple eddy fronts and, notably, remained within a single anticyclone (ACE-01) for
14 days.

While the eddies in these cases are indeed highly variable (with different ages, intensities,
buoyancy structures, etc.), they exhibit a consistent pattern: the lateral buoyancy gradients (bx)
and mesoscale strain are of a similar order of magnitude across them (Figs. 2, S3, S4).
Consequently, the Seaglider recorded a clear pattern during these transects: the frontal regions at
the eddy edges consistently feature strong bx and VHT, whereas the eddy interiors are
comparatively quiescent with much weaker gradients, w , and VHT. This demonstrates that our
observational analysis can capture systematic patterns within these highly variable eddies, thus
providing a sound basis for the subsequent composite analysis.

To provide a more intuitive and transparent justification for our composite methodology, we have
also introduced a figure illustrating single-eddy composites (Fig. S13, also shown as Fig. R6
below). For this, we selected eddies from our case studies—ACE-02 and ACE-04 from the
sg559-T2 transect (Fig. S3) and the long-duration ACE-01 from the sg561-T3 transect (Fig.
S4)—and projected their data individually onto the normalized eddy-centric coordinate system
(Fig. S13a-d and S13e-h, respectively). The results demonstrate that strong signals of bx and VHT
originating from the eddy edges are consistently mapped to the annular region at a normalized
radius of $r \approx R$. Our method does not perform any rotational alignment, thereby preserving the
original azimuthal orientation of each feature. This figure clarifies that while our composite is a
statistical average, it clearly shows physically relevant signals (such as submesoscale signals at
eddy edges) rather than "smoothing them away," providing more interpretable, case-specific
support for our composite framework.

Furthermore, in contrast to the VHT associated with mesoscale processes within the eddy interior
($<R$), the strong VHT driven by submesoscale dynamics occurs preferentially at the eddy edge.
This submesoscale-driven heat transport is predominantly upward for both cyclones and
anticyclones. Consequently, a primary finding highlighted by our composite analysis is this
fundamental difference in VHT patterns between the eddy edge and non-edge regions (i.e., the

eddy interior and exterior). We understand the reviewer's concern regarding eddy structure and
 heterogeneity, and as shown in both our Seaglider results (Figs. 2, S3, S4) and model outputs (Figs.
 S5, S6), the strongest VHT signals ($> 1000 \text{ W m}^{-2}$) are indeed co-located with the strongest strain
 and lateral buoyancy gradients ($|b_x|$), i.e., at the fronts.

We would also like to add two points: (1) The colored shaded areas in our manuscript's radial
 distribution plots (Fig. 3**b,c,f,g,j,k**) represent the standard error. This error is small at the eddy
 edge ($r \approx R$). This illustrates that the strong signal is a robust feature across the ensemble, not an
 artifact of a few extreme events. (2) Our analysis in Fig. 5 employs a conditional composite,
 where eddies are grouped by the intensity of the geostrophic strain at their edge. The results show
 a positive correlation: eddy groups with stronger strain exhibit stronger VHT at their edges.

This study is the first to use such a long-term, continuous, observational dataset to confirm that
 submesoscale processes at eddy edges drive intense VHT in this high-latitude region. This finding
 was made possible by applying composite analysis to extensive glider datasets. These substantial
 revisions, including the new case studies and methodological clarifications, have significantly
 improved the manuscript. We thank the reviewer again for the valuable comments.

 **Fig. R4** | **a** FSLE, **b** SLA superimposed on the Seaglider track from 6 September 2012 to 10
 October 2012. The glider's vertical section of **c** buoyancy, **d** lateral buoyancy gradient (b_x), **e**
 vertical velocity (w). **f** The glider's vertical section of vertical heat transport (VHT), with positive
 (negative) values denoting upward (downward) heat transport. The mixed layer depth (MLD) is
 shown by a thick black line in **c-f**. In panels **a** and **b**, red and blue dots indicate where the

Seaglider traversed the edges of four anticyclonic eddies (ACEs) and one cyclonic eddy (CE),
 respectively, corresponding to the same color-coded positions along the Seaglider track displayed
 above panels c-f.

 **Fig. R5** | a FSLE, b SLA superimposed on the Seaglider track from 6 January 2014 to 17 February
 2014. The glider's vertical section of c buoyancy, d lateral buoyancy gradient (b_x), e vertical
 velocity (w). f The glider's vertical section of vertical heat transport (VHT), with positive
 (negative) values denoting upward (downward) heat transport. In panels a and b, red and blue dots
 indicate where the Seaglider traversed the edges of three ACEs and two CEs, respectively, with
 the glider remaining within one anticyclonic eddy (ACE-01) for 14 days (marked by yellow dots).
 These dots correspond to the same color-coded positions along the Seaglider track displayed
 above panels c-f.

**Fig. R6 | Case study of Seaglider observations in normalized eddy-centric coordinates. a-d**

Seaglider sg559 transect T2 through ACE-02 and ACE-04 (see black dashed boxes in Fig. R4b): **a**

buoyancy, **b** lateral buoyancy gradient (b_x), **c** vertical velocity (w), and **d** vertical heat transport

(VHT). The circles and triangles denote the tracks of ACE-02 and ACE-04, respectively. **e-h**

Seaglider sg561 transect T3 through ACE-01 (see black dashed box in Fig. R5b): **e** buoyancy, **f**

lateral buoyancy gradient (b_x), **g** vertical velocity (w), and **h** VHT. All presented variables are

averaged over the upper 500 m. R represents the normalized radius of the eddy.

**“Warm Rings in Mesoscale Eddies in a Cold Straining Ocean”**

Huizi Dong, Meng Zhou, James C. McWilliams, Roshin P. Raj, Francesco d’Ovidio, Ilker Fer,
Lixin Qu, Bo Qiu, Lia Siegelman, Zhengguang Zhang, Walker O. Smith, Jr., Ann Kristin
Sperrevik

**Note:** Reviewer’ comments are in italic font; authors’ response comments are in normal font.
Revisions in the revised manuscript are highlighted.

**REVIEWER COMMENTS**

***Reviewer #4 (Remarks to the Author):***

*The manuscript titled “Warm Rings in Mesoscale Eddies in a Cold Straining Ocean” provides a*
*well written report on submesoscale heat transport in the Lofoten Basin and reveals novel insights*
*into the dynamical regimes present. The results have been presented in a clear and concise*
*manner, and the analysis is of a sound standard. The primary purpose of my review is to provide*
*additional guidance on the whether concerns raised by Reviewer 3 have been sufficiently*
*addressed. I believe these points have been largely resolved, but minor deficiencies remain in the*
*discussion of smearing and aliasing effects. My review reflects on the response to each of*
*Reviewer 3’s points in turn, followed by two additional (minor) suggestions.*

**Response:** We thank the reviewer for their positive and constructive feedback. We are grateful for
their assessment that the previous concerns have been largely resolved and appreciate the clear
suggestions for the remaining minor but important issues. We have addressed all the points below.

*Point 1 – The authors raise valuable points in response to the assertion that the manuscript lacks*
*novelty and I believe the paper is of significance to the field. However, the narrative in some*
*places leads the reader to conclude that the present study is the first of its kind, when the headline*
*result is building on that of Siegelman et al. (2020). Although not critical for publication, some*
*minor changes to the text may help differentiate the manuscript’s novel findings from that of*
*existing literature. My main recommendations are that (1) the study’s isolation of the cyclonic and*
*anticyclonic contributions to submesoscale vertical heat transport is made clearer on lines*
*106-108 and (2) the framing of the text between lines 284-290 is refined to acknowledge that*
*submesoscale heat fluxes in regions of strain has been identified in previous studies.*

**Response:** Following the reviewer’s suggestions, we have revised the text to more explicitly
highlight the isolation and comparison of VHT in cyclonic and anticyclonic eddies. The revised
text now reads (lines 106-111):

“Building on this intriguing observation, a key novelty of our study is the isolation and
comparison of VHT in both cyclones and anticyclones. We investigate how deep-reaching
submesoscale motions drive a systematic upward heat flux along their respective edges, providing
a dynamical explanation for the "warm ring" phenomenon in a comparative context that has been
largely overlooked.”

We have also revised the Discussion section to better acknowledge previous work on strain-driven
heat fluxes. The new text is as follows (lines 284-292):

“However, while these lateral transport processes remain important, previous work has identified
enhanced submesoscale heat fluxes in strain-dominated regions²³. Building upon this, our study
reveals that within mesoscale eddies, vertical transport is largely governed by submesoscale
ageostrophic motions along both cyclonic and anticyclonic edges (Supplementary Information and
Figs. S8–S9). Specifically, the continuous vertical structures of w are primarily observed in the
strain-dominated areas around eddies, serving as a crucial pathway for the vertical exchange of
heat and energy between the ocean interior and the surface.”

*Point 2 – The authors have made considerable efforts to respond to the Reviewer 3’s concern on*
*data contamination. In my view, their expansion of the analysis to compare spectral*
*decompositions and analysis of the impact of isopycnal filtering sufficiently addresses the*
*reviewers concern around the lack of transparency in qualifying the use of glider observations for*
*the manuscript’s purposes. Results showing that fine-scale features remain in the buoyancy*
*gradient and vertical velocity fields upon isopycnal filtering is sufficient indication that these are*
*associated with submesoscale dynamics. The authors have taken on board the reviewer’s*
*suggestions, but in the process have omitted a key distinction between smearing and aliasing that*
*would aid their case. For example, line 305 quotes “Doppler aliasing” when this should be*
*“Doppler smearing”. As outlined by Rudnick et al. (2011), aliasing is a function of sampling rate*
*and Doppler smearing is associated with the speed that the observing platform travels through the*
*water. As a minimum, I would recommend that a distinction is made in the text between the source*
*of aliasing and smearing, and that the authors are careful not to use the terms interchangeably.*
*Furthermore, though not necessary for publication, it could benefit the manuscript to cite the*
*relevant length and spatial scales associated with these effects (e.g., Rudnick et al., 2011). It is*
*possible to estimate the spatial- and temporal-scales upon which smearing and aliasing act, and*
*the sampling intervals cited in the supplementary material indicate there may be limited aliasing*
*at the submesoscale in the observations in question.*

**Response:** We thank the reviewer for pointing out the need to distinguish clearly between Doppler
smearing and aliasing. Accordingly, we have replaced the term “Doppler aliasing” with the correct
“Doppler smearing” (lines 309–310 of the revised version).

We have also updated the “Analysis of the Seaglider data” section (now lines 103-110 in the
Supplementary Information) to provide precise definitions of both terms and to cite Rudnick et al.
(2011). The revised text now reads:

“A key challenge in using glider data for submesoscale studies is potential contamination from
high-frequency oceanic phenomena like internal waves, which manifests as two distinct effects:
Doppler smearing and aliasing. Doppler smearing is a consequence of the glider's finite speed
through the water (~0.2 m/s) and manifests as a smearing of the true wave number and
frequency¹⁰. In contrast, aliasing is a function of the sampling rate, which causes variability from
higher wave numbers to be folded into the resolved lower wave numbers¹⁰.”

*Point 3 – The authors have performed the additional analysis requested by Reviewer 3 regarding*
*the composite analysis and their concerns have been addressed to a satisfactory level. Despite*
*significant non-uniformity in eddy structure, the composite analysis appears to provide a robust*
*measure of the alignment between vertical heat transport and strain.*

**Response:** We appreciate the reviewer for acknowledging our composite analysis.

***Additional Suggestions:***

*The sentence starting “The rotational flow...” on lines 279-282 is difficult to interpret. As far as I*
*can tell, the intention is to convey that eddies can advect water masses. Since this is already stated*
*in the previous sentence, I would suggest this sentence is removed. In addition, Reference 35 does*
*not appear to have relevance to the claims made and, should the sentence remain, I would suggest*
*reviewing whether this citation is appropriate.*

**Response:** We agree with the reviewer and have removed this sentence from the revised version.

*Line 287 currently reads “eddy eddies”. Should this be “eddy edges”?*

**Response:** The typo has been corrected: “edgies” is now “edges” in line 288 of the revised
version.

*I hope this feedback is received as constructive and I look forward to seeing the work published,*
*should that be the final decision.*

Second Review of 'Warm Rings in Mesoscale Eddies in a Cold Straining Ocean' by Huizi Dong et al.

At the request of Nature Communications, I have conducted a second review of the revised manuscript by Dong et al. I would like to first express my appreciation for the authors' thorough and thoughtful responses to my initial comments. Overall, I am satisfied with the revisions, and I do not have any major concerns that would warrant substantial additional changes. However, I would like to offer the following two comments for the authors' consideration:

1. TS Distribution & Salinity Contribution

I appreciate the authors' inclusion of salinity transects and T–S diagrams, as well as the quantitative assessment of the respective contributions of temperature and salinity to density and geostrophic flow.

The vertical distributions of temperature and salinity along Transect 1 (Fig. R1b–c) illustrate the typical hydrographic structure of the region, where temperature variations (2–4 °C) are larger than salinity variations (<0.1–0.2 psu), and thus have a greater influence on local density.

I agree with the authors that, within the parameter range considered (temperature = 4–8 °C, salinity = 35.2–35.4), temperature variations predominantly govern the density structure.

As an additional suggestion, rather than qualitatively describing this point with multiple phrases, I recommend introducing the density ratio, $R\rho = (\alpha \Delta T) / (\beta \Delta S)$, which offers a concise and quantitative way to demonstrate the dominant role of temperature anomalies.

Using typical values:

- $\alpha \approx 2 \times 10^{-4} \text{ } ^\circ\text{C}^{-1}$

- $\beta \approx 8 \times 10^{-4} \text{ psu}^{-1}$

the density ratio is approximately $R\rho \sim 5$, clearly indicating that temperature changes have a much greater impact on density than salinity variations under these conditions. This would help clarify the physical basis of the authors' interpretation and improve the overall readability of the manuscript.

2. Geostrophic Velocity Resolution

Thank you for the authors' explanation and response regarding the concern about the resolution of satellite SSH data in detecting mesoscale eddies. I particularly appreciate the quantitative assessment of the typical size of these eddies.

While the local deformation radius is around 10 km, our analysis focuses on these mesoscale eddies with 30–50 km radii that are well resolved by the available altimetry data.

However, I believe the logic provided in the response is flawed. The key issue is not how many local deformation radii fit into an eddy of 30–50 km radius, but rather whether the satellite SSH data have sufficient resolution to adequately resolve such features.

At 70°N, the meridional resolution of a 1/4° gridded product corresponds to approximately 27.8 km, meaning that an eddy may only be represented by one or two grid points in the meridional direction—clearly limiting the accuracy of structure detection.

On the other hand, the zonal resolution is finer, approximately 9.5 km, allowing for about 3 to 5 grid points across the eddy in the zonal direction. This suggests that, at least in the zonal direction, the horizontal velocity structure of the eddy can be reasonably resolved.

Assuming the eddies are roughly circular in shape (even if somewhat deformed), I recommend that the authors briefly clarify in the manuscript that resolving eddy structure in the zonal direction may be sufficient to enable satellite-based eddy detection in this region.